# Using eDNA techniques to find the endangered big-headed turtle (*Platysternon megacephalum*)

**Ivan P. Y. Lam, Yik-Hei Sung, Jonathan J. Fong** *

Science Unit, Lingnan University, Hong Kong, China

* jonfong@ln.edu.hk

**Data Availability Statement:** All relevant data are within the paper and its Supporting Information files.

**Funding:** The sources of funding of the work submitted are as follows: 1. Research Grants

## Abstract

We evaluate the efficacy of environmental DNA (eDNA) techniques to locate wild populations and estimate the population size of the endangered big-headed turtle (*Platysternon megacephalum*) in Hong Kong. The results from this study are important for identifying priority sites for protection and further research. Additionally, we assess the impact of two environmental variables (temperature and pH) on eDNA quantity. We surveyed 34 streams for three years, sampling four times each year. Four new populations were first identified with eDNA analysis, and then verified by field surveys. Our multi-year survey highlights that eDNA detection can be inconsistent over time, even in streams with known populations. There was no significant relationship between eDNA quantity and the environmental variables tested. Lastly, our results suggest that eDNA methods remain promising to estimate population size, since number of positive detections were positively correlated with population size in streams with known populations. We conclude that eDNA methods are powerful, but care must be taken when interpreting field results as they are affected by species ecology and environmental conditions.

## Introduction

The big-headed turtle (*Platysternon megacephalum*) is distributed across East and Southeast Asia (China, Thailand, Vietnam, Cambodia, Laos, and Myanmar) [1]. Populations of *P. megacephalum* across its range have declined drastically due to severe hunting pressure [2–5]. This decline was highlighted in two studies conducted in South China, where they found only 16 individuals in over 4000 trapping days [6, 7]. This species is currently listed on CITES Appendix I [8] and classified as "Critically Endangered" on the IUCN Red List of Threatened Species [9] with recommendations [10].

While *P. megacephalum* has largely been extirpated across its geographical range, a remnant population exists in Hong Kong, providing a unique opportunity for research and conservation. However, work on *P. megacephalum* is hindered by incomplete knowledge of where wild individuals persist due to its secretiveness and rarity. Environmental DNA (eDNA) refers to DNA deposited in the environment (such as water, soil, and air) originating from the target species, and eDNA-based techniques provide a tool to assess the presence of a target species

Council of Hong Kong (Early Career Scheme #23100216), from The Government of Hong Kong Special Administrative Region; received by J.J.F. (URL: https://www.ugc.edu.hk/eng/rgc/) 2. Environment and Conservation Fund (#ECF2017-04), from The Government of Hong Kong Special Administrative Region; received by Y.H.S. (URL: https://www.ecf.gov.hk/en/home/index.html) 3. Ocean Park Conservation Foundation Conservation Fund (#RP01.1718), from Ocean Park Conservation Fundation Hong Kong; received by Y.H.S. (URL: https://www.opcf.org.hk/en/) 4. Croucher Foundation Chinese Visitorship (#870026), from Croucher Foundation; received by J.J.F. (URL: https://croucher.org.hk/funding/enabling-the-exchange-of-ideas-between-hong-kong-scientists-and-their-counterparts-in-mainland-china-and-overseas/croucher-chinese-visitorships) The funders had no role in study design, data collection and analysis, decision to publish, or preparation of the manuscript.

**Competing interests:** The authors have declared that no competing interests exist.

without observing it [11]. Methods have been developed to detect the eDNA of a wide range of taxa [12–17]. Researchers have also tested the prospect of using eDNA approaches to estimate abundance, with some studies successful [16, 18–20] and others unsuccessful [21–24]. Recent achievements include locating a new population of the endangered Yamoto salamander (*Hynobius vandenburghi*) [25], differentiating populations of the harbour porpoise (*Phocoena phocoena*) [26], and potential use in population genetics studies [27, 28]. In this study, we test whether eDNA techniques can be applied to help locate, study, and conserve *P. megacephalum*.

In this study, we use a validated eDNA assay [29] to clarify the distribution of *P. megacephalum* across Hong Kong. Since *P. megacephalum* is a highly aquatic species with rare terrestrial movement [4], eDNA-based stream surveys are expected to assist in locating the species. The aims of our study are to evaluate the (1) usefulness of eDNA techniques in detecting lotic turtle populations, (2) impact of two environmental variables (temperature and pH) on eDNA quantity, and (3) effectiveness of eDNA data in abundance estimation. We conclude by discussing the potential factors that influence eDNA detection (eDNA degradation, production, transport, and dilution). For *P. megacephalum* in Hong Kong, the results from this study are important for identifying priority sites for protection and further research.

## Materials and methods

### Study site

The study was conducted in the Hong Kong Special Administrative Region, China (22˚09'–22˚37'N, 113˚50'–114˚30'E). Hong Kong has a subtropical climate characterized by cool, dry winters ("dry season", November to February) and hot, humid summers ("wet season", May to September) separated by mild autumns and springs [30]. A total of 34 streams with suitable habitat for *P. megacephalum* (rocky streams with fast flowing, clear water in secondary forest) were selected for eDNA water sampling. Streams were divided into three categories: (1) known populations based on long-term trapping surveys (KP; 8 streams), (2) historical records of presence with the latest report more than 10 years ago (HR; 9 streams), and (3) suitable habitat but unknown status (UN; 17 streams). Exact survey locations are not disclosed to protect populations of *P. megacephalum*, as this species is listed as Endangered (EN) on the IUCN Red List [9] and Appendix I of CITES [8], and subject to poaching [2–5].

### Sample collection and processing

We collected water samples over an approximately three-year period (November 2016–August 2019). For each year, we collected samples twice for each wet season (in May and August) and dry season (in November and February). A total of 12 water samples were collected from each stream (4 samples/year × 3 years = 12 samples). One liter of water was collected from the water column [13, 31–33] using a sterile 50-mL conical tube. At the same time, water temperature and pH were measured using a waterproof temperature/pH meter (Eutech pHTestr 30). Water samples were collected in sterile Whirl-Pak® sample bags (Nasco #B01027; WI, USA) and filtered using 0.45 μm pore size, cellulose nitrate filters (Nalgene analytical test filter funnel #145–2045; Thermo Fisher Scientific Inc; Waltham, MA, USA) either on site or in the laboratory within six hours. Filters were removed from the funnel using sterile forceps, with gloved hands, and stored in 95% ethanol at -20˚C until DNA extraction was performed.

Prior to each lab work step, we sterilized laboratory benches and equipment with a 10% bleach solution for at least 5 minutes. The filters were divided into two pieces to provide two opportunities for DNA extraction. DNA was extracted using a DNeasy Blood & Tissue Kit and QiaShredder (Qiagen GmbH; Hilden, Germany) following a validated protocol [34]. DNA was

eluted using 100 μL of TE buffer and stored at -20˚C. The DNA concentration of each sample was measured using a Qubit 3 Fluorometer with the dsDNA HS Assay Kit (InvitrogenTM; Thermo Fisher Scientific Inc; Waltham, MA, USA). Samples were used in subsequent quantitative PCR (qPCR) analyses if DNA quantification recovered a positive value greater than the detection limit of the fluorometer (> 0.2 ng/μL). If DNA was not detected during quantification, DNA was extracted again from the second half of the filter, and amplified independently in qPCR without combining with the first extraction. Negative controls were included for all laboratory work to identify any contamination throughout sample processing (one for each set of twenty-three samples of DNA extractions, one for each set of twenty-three samples for DNA quantification, and two or three for qPCR in a 96-well plate).

## Quantitative PCR

qPCR reactions were performed on a StepOnePlus Real-Time PCR System (Applied Biosystems; Thermo Fisher Scientific Inc; Waltham, MA, USA). We followed the qPCR conditions of a *P. megacephalum* specific assay targeting the *ND4* region [29], of which the analytical sensitivity at 10 copies/μL was 0.95, indicating reliable detection of eDNA down to 10 copies/μL. A validation checklist of the selected qPCR assay following [35] can be found in S1 Table. We summarize the protocol herein. Each qPCR reaction included 1 μL eDNA template, 5 μL TaqMan® Environmental Master Mix (Thermo Fisher Scientific Inc; Waltham, MA, USA), 900 nM of each primer, 250 nM of probe, and enough autoclaved Milli-Q® water to make a final volume of 10 μL. The TaqMan® Exogenous Internal Positive Control (IPC) (1 μL IPC-Mix and 0.2 μL IPC-DNA following manufacturer's protocol) was included to test for inhibition within each qPCR reaction. Negative controls (two or three for each 96-well plate) were included in all tests. Thermal cycler conditions were as follows: 50˚C for 2 min, then 95˚C for 10 min, followed by 40 cycles of 95˚C for 15 s and the 60˚C for 1 min. To quantify absolute concentration of eDNA, we employed a four-level standard curve (1,000,000 copies/μL, 100,000 copies/μL, 10,000 copies/μL, and 100 copies/μL), using a synthetic gene containing primer and probe binding sites (Tech Dragon Limited; Hong Kong). Serial dilution of the synthetic gene was done one week prior to qPCR to minimize contamination. qPCR fluorescence signal, threshold value and $C_q$ value were calculated using the StepOne™ software v.2.3 (Applied Biosystems; Thermo Fisher Scientific Inc; Waltham, MA, USA).

For each water sample, we ran multiple qPCR reactions to determine the presence/absence of *P. megacephalum* eDNA [36]. Here, we define a few terms to distinguish between the results of a single qPCR reaction and the sample as a whole (two to five individual qPCR reactions). For a single qPCR reaction, we use the term "amplified" and similar terms (e.g., amplification) when qPCR amplification was observed above the default threshold value, and the qPCR curve exhibited a typical sigmoidal shape. For the sample as a whole, it was categorized as either positive detection, negative detection, or uncertain to indicate the presence of *P. megacephalum*, based on the combined result of all qPCR replicates performed on the sample (details below).

Fig 1 is a diagram of the assay workflow, separated in two phases. For the first phase, each DNA template was tested in duplicate. A sample was considered to have positive detection of *P. megacephalum* if both replicates amplified, and negative detection when both replicates did not amplify. A sample with 1/2 amplified replicates was tested again in the second phase. In the second phase, an additional three replicates were run for each sample. If two or three of these new replicates amplified, the sample was considered to have positive detection of *P. megacephalum*. If none of the three replicates showed amplification, the sample was considered to have negative detection of *P. megacephalum*. Lastly, if one of these new replicates amplified, the sample was considered as uncertain.

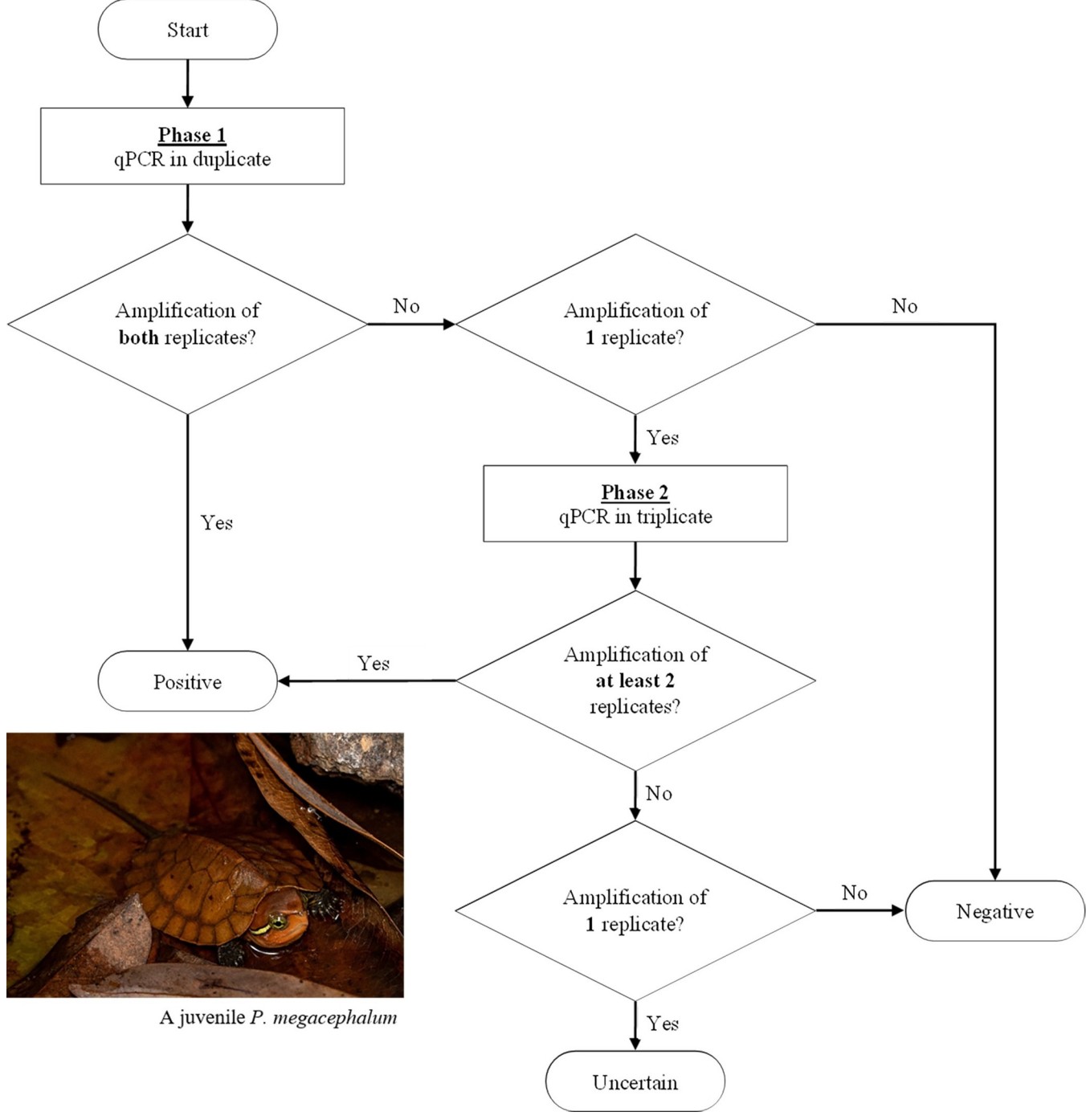

**Fig 1. A diagram of the two-phase workflow to determine qPCR results.**

## Turtle surveys

These field survey data serve as a way to crosscheck the eDNA-based survey results. Trapping at KP sites is part of a long-term population monitoring program started in 2009. Since then, trapping surveys have been conducted one to two times a year in each stream, using baited hoop traps following protocol in [37]. For all HR and UN streams, we conducted at least one

field survey at each site, using trapping and/or active searching. When trapping, we set at least 10 traps along the stream where we took water samples for eDNA analysis. For active searching, we walked along the stream looking for *P. megacephalum* using headlamps at night. Permits to possess and deploy traps and temporarily retain freshwater turtles were approved by the Agriculture, Fisheries and Conservation Department, HKSAR [(86) in AF GR CON 09/51 Pt. 7)].

### Exploration of eDNA data

We explored three aspects of the eDNA data using correlation analysis: (1) usefulness of eDNA techniques in detecting lotic turtle populations, (2) impact of two environmental variables (temperature and pH) on eDNA quantity, and (3) effectiveness of eDNA data in abundance estimation. Shapiro-Wilk tests were used to test the normality of data for eDNA quantity, number of eDNA amplified samples, mean capture rate, and environmental variables. The only normally distributed ($p < 0.05$) variable was the number of eDNA amplified samples. We selected Kendall's correlation analysis to evaluate the relationship between parameters. Statistical analyses and graphical representations were performed using R version 4.0.3 [38] and RStudio version 1.2.5042 [39]. For all correlation analyses, we subsampled the dataset to include samples with qPCR efficiency 90–110% [40], eDNA concentration above the limit of quantification (100 copies/μL), and KP streams. KP streams were selected because the presence of *P. megacephalum* is known and detection of eDNA is expected to be more consistent at these sites. Additionally, we compared results of two datasets: (1) positive detection and uncertain samples, and (2) positive detection samples only.

First, we assessed the usefulness of using eDNA to detect lotic turtle populations, by comparing the number of samples with positive detection (maximum 12 samples/site) to capture rate (total number of turtles caught/total number of traps set). We used capture rate as an estimate of population size. For this analysis, we restricted the capture rate data to the trapping/visual surveys done during the study period. Next, we explored how environmental variables affect eDNA quantity by analyzing the relationship between temperature or pH and eDNA quantity. We used two site-based eDNA quantity datasets: stream KP5 only (the one stream with near-continuous eDNA detection) and all KP streams (n = 8). Lastly, we assessed the effectiveness of eDNA data for abundance estimation, by comparing eDNA quantity to capture rate. To minimize the impact of comparing asynchronous data from eDNA and trapping surveys, we only included data when a trapping and eDNA survey were conducted within a month of each other. Based on this criterion, data from 35 trapping/eDNA surveys were included in the analysis. As with the environmental variable analysis, eDNA quantity is based on two datasets—stream KP5 only and all KP streams.

## Results

### Quantitative PCR

A total of 408 water samples (34 locations × 3 years × 4 samples/year) were collected during the study. DNA was detected in all DNA extractions from the first half of the filter. Details and qPCR results for all replicates are found in S2 Table (qPCR efficiency: 96.99% ± 5.19; $R^2$: 0.957 ± 0.03). No eDNA samples exhibited inhibition, and none of the negative controls exhibited qPCR fluorescence signal. All amplified water samples were above the limit of detection (10 copies/μL) as validated in [29], while all negative water samples yielded no qPCR fluorescence signal. Based on our qPCR testing approach (Fig 1), there was amplification of *P. megacephalum* eDNA in 37 samples: 24 samples were classified as positive detection and 13 samples were classified as uncertain. These 37 samples with amplification were from six KP, three HR,

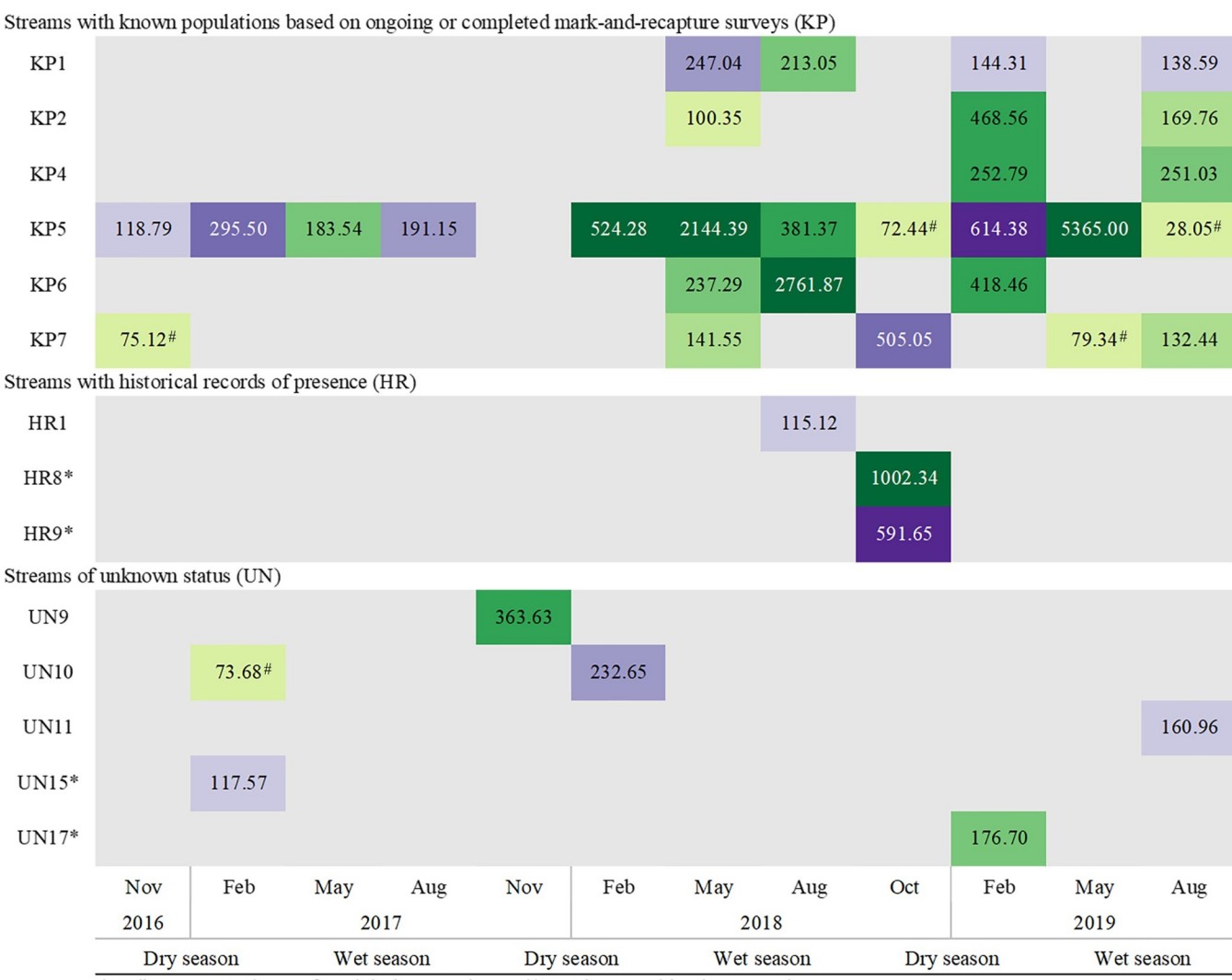

**Fig 2. Heat map illustrating the concentration of eDNA detected in water samples from November 2016 to August 2019.** Each depth of color indicates a percentile of concentration level (interval = 0.2). A darker color represents a higher concentration level for positive detection (in green) and uncertain detection (in purple). Grey represents samples with no amplification, and categorized as negative detection.

and five UN streams (Fig 2). Fig 2 is a heat map illustrating the temporal change of eDNA concentration over the study period from streams with eDNA amplification. The remaining 371 samples were classified as negative detection. A summary of amplified samples across the three categories of streams can be found in Table 1.

Most of the samples with amplification (76%; 28 samples) were from KP streams. Of the eight total KP streams, only one (KP5) showed consistent eDNA amplification across the three-year period (11 out of 12 samples). Two KP streams had negative detection for all 12 samples during the study period. For the remaining five KP streams, there was occasional eDNA amplification (two to five samples with amplification), with samples with amplification being more frequent after May 2018.

Three of the samples with amplification (~8%) came from three HR streams. No amplification was detected in any of the 12 eDNA samples from the six remaining HR streams. The six

**Table 1. A summary of amplified samples across the three stream categories: Known Population (KP), Historical Record (HR), Unknown (UN).**

|  | KP | HR | UN | Total |
|---|---|---|---|---|
| Positive | 20 | 1 | 3 | 24 |
| Negative | 69 | 105 | 198 | 372 |
| Uncertain | 7 | 2 | 3 | 12 |
| Total | 96 | 108 | 204 | 408 |

remaining samples with amplification (~16%) came from five UN streams. No amplification was detected from any of the 12 eDNA samples from the remaining 12 UN streams. Among the HR and UN streams with amplification, most had only one sample with amplification throughout the three-year study period.

## Turtle surveys

Table 2 summarizes the data from trapping surveys of KP streams during the study period, while capture rate corresponding to each eDNA sampling season can be found in S3 Table. There were two KP streams (KP3, KP8) that were categorized as negative detection in all samples throughout the study period. These two streams had the two lowest mean trapping rates (KP3 = 0.00, KP8 = 0.03) of the KP streams, likely indicating relatively small populations. Active searching and trapping outside of the eDNA study period verified the presence of *P. megacephalum* in both of these streams. Of the HR and UN streams with amplified *P. megacephalum* eDNA (three HR, five UN), the presence of *P. megacephalum* was either unconfirmed (HR8, HR9, UN15, UN17), or confirmed by trapping (UN10, UN11) or active searching (HR1, UN9). For the remaining 18 streams with negative eDNA detection (six HR streams, 12 UN streams), no turtles were caught or seen.

## Exploration of eDNA data

The results were the same for both datasets (positive and uncertain samples, positive only). Here we detail the statistical results from the dataset of positive and uncertain samples. First, for assessing the usefulness of eDNA methods to detect lotic turtle populations, we found a significant, positive correlation ($r_s$ = 0.718, $p$ = 0.02) between the number of samples with amplified eDNA and mean capture rate (Table 2). In other words, sites with a higher number of positive detections had a higher mean capture rate and likely larger, more stable populations. Next, for assessing the impact of two environmental variables (temperature and pH) on eDNA quantity, no significant relationship was found for both site-based eDNA quantity datasets

**Table 2. Summary of trapping surveys throughout the eDNA sampling period at streams with known populations (KP).** There were positive sightings of *Platysternon megacephalum* using active searching at all streams.

| Stream | Number of surveys | Mean capture rate |
|---|---|---|
| KP1 | 9 | 0.14 |
| KP2 | 3 | 0.18 |
| KP3 | 5 | 0.00 |
| KP4 | 1 | 0.10 |
| KP5 | 8 | 0.56 |
| KP6 | 3 | 0.27 |
| KP7 | 7 | 0.67 |
| KP8 | 1 | 0.03 |

(KP5 only: $r_s$ = -0.056, $p$ = 0.814 and $r_s$ = -0.294, $p$ = 0.212, and all KP streams: $r_s$ = 0.043, $p$ = 0.619 and $r_s$ = -0.277, $p$ = 0.116 for temperature and pH respectively]. Lastly, for assessing the viability of abundance estimation using eDNA quantity, there was no significant relationship for the dataset of KP5 only ($r_s$ = -0.036, $p$ = 0.90), and for the dataset of all KP streams ($r_s$ = 0.267, $p$ = 0.056).

## Discussion

We demonstrate the usefulness of eDNA-based surveys to locate wild turtles in lotic systems by using a previously developed qPCR assay for *P. megacephalum* [29]. We targeted three types of streams: (1) known populations based on long-term trapping surveys (KP), (2) historical records of presence (HR), and (3) suitable habitat but unknown status (UN). For the KP streams, the detection results varied; of the 12 samples from a single stream (4 samples/ year × 3 years), eDNA detection between streams ranged from nearly continuous (11 of 12 samples) to no detection (no positive samples), with no clear variation based on time of year. Focusing on these streams with known populations, the number of eDNA detections was positively correlated with mean capture rate, suggesting that sites with larger populations have higher probability of eDNA detection.

In eight streams without known populations, eDNA of *P. megacephalum* was occasionally detected (one to two positive samples). Three of these streams had historical records (HR streams), while the other five streams have no previously confirmed turtle populations (UN streams). We carried out follow-up surveys (trapping and/or active searching) at these streams, and turtles were found in four streams (HR1, UN9, UN10, and UN11). We raise several possible explanations of these results, of which we cannot currently distinguish between. It should also be noted that with positive results from several streams, different explanations may be needed for each situation.

First, these occasional positive results could be false positives. Template concentration was found to affect primer specificity [41], although this factor had no significant effect in another study [42]. More specifically, a high concentration of non-target template may cause non-specific qPCR amplification [41]. Since DNA concentration for our samples was generally low (average eDNA concentration of amplified samples collected in our study is 2.70 ng/μL), we do not expect such false positives to be a major factor in our study. The remaining potential explanations assume that the results are true positives—the qPCR assay is species-specific, and the samples contain eDNA of *P. megacephalum*. We discuss these potential explanations below in the sections about eDNA degradation, production, transport, and dilution.

Estimation of population size has been raised as a potential use of eDNA-based surveys. Since we had eDNA quantity and capture rate data from the same streams, we had an opportunity to assess the viability of estimating population size based on eDNA results. There was a significant, positive correlation between eDNA quantity and capture rate from streams with known populations, but no such relationship when focusing on the one stream with continuous eDNA detection (KP5). These results suggest that eDNA methods remain promising to aid in estimating population size or biomass, but need to be refined based on differences in the habitat and species ecology. Although the quantity of eDNA should increase with more individuals, eDNA detected with qPCR assays may not be proportional to the number of individuals due to eDNA degradation, production, transport [43], and dilution. These processes are influenced by environmental factors and cause eDNA quantity to vary, making abundance estimation difficult. We discuss these four processes in detail, as well as the issue of false negatives and limitations to our study.

## eDNA degradation

Researchers have attempted to clarify how environmental factors affect eDNA degradation in water. Two major variables studied are temperature and pH. Multiple studies indicated that higher water temperature increases eDNA degradation rate [43–47], while other studies did not find any significant relationship [16, 48, 49]. One caveat is that temperature may also increase eDNA concentration, which we discuss in the eDNA production section below. pH may also facilitate eDNA degradation [43, 46]. We tested for a relationship in the single stream with relatively consistent eDNA detection (KP5), and no significant relationship was found. Our result is similar to the results of another study [48] that found eDNA did not vary significantly with pH and water temperature (and dissolved oxygen and turbidity), when using eDNA methods to study the lake trout (*Salvelinus namaycush*). Although we tried to control for the effects of other variables by focusing on a single stream, temporal variation in eDNA production and transport may have influenced our results. To advance our knowledge of environmental factors on eDNA degradation, we suggest controlled laboratory experiments, followed by mesocosm-type experiments in the field.

## eDNA production

eDNA production can be influenced by environmental factors and life stage of the organism. An increase in water temperature can increase eDNA production, due to an increase in an animal's metabolism [49, 50]. This phenomenon is also expected in poikilothermic turtles, but has yet to be studied. A counterpoint to this is the "shedding hypothesis" [51], which suggested turtles would shed less eDNA compared to other animals, as they are covered with keratinized integument.

The life stage of an organism has also been documented to affect eDNA production [52–54]. A significant increase in eDNA quantity during the breeding season of the target species was documented in [23]. The breeding season of *P. megacephalum* is from May to July, with hatchlings emerging in October [55]. At the same time, the increase of eDNA in the breeding season may be less significant to species that live in low-density, as the effect would be easily masked by multiple environmental factors that influence eDNA quantity. Unfortunately, our sampling was not frequent enough to detect changes during the breeding season. We encourage researchers to address such questions for their specific study organism, in the hope that combined datasets can give us a general view of eDNA production.

## eDNA transport

The transport of eDNA in an environmental system needs to be taken into consideration, especially for lotic systems. Unlike lentic systems where eDNA distribution is highly localized in space and time [56], a lotic system with a fast flowing stream can cause eDNA distribution to vary in an unknown way. A controlled experiment of a lotic system showed that eDNA is transported unevenly, and eDNA retention and resuspension involves the benthic substrate [57]. As benthic sediment and hydrology differ between streams, each study site will vary in the way eDNA is transported [57]. Additionally, eDNA in lotic systems has been demonstrated to travel long distances, with no significant decrease of DNA concentration for over 1.7 km [58] and even up to 9.1 km [59]. Transportation of eDNA raises the potential of the target species being located upstream of where the water sample was collected. In our study, eDNA was detected in four streams with unconfirmed presence of *P. megacephalum*, and we believe the assay may be detecting eDNA from turtles located further upstream in areas inaccessible for trapping and active searching.

## eDNA dilution

Dilution caused by high stream flow affects eDNA concentration through multiple mechanisms, potentially with opposite outcomes. eDNA dilution may reduce detection probability [60]. In the wet season, eDNA is expected to be diluted via higher water volume and faster water flow, hence reducing eDNA detection probability or causing false negatives [60, 61]. On the other hand, fast water flow may lead to false positives by resuspending historical eDNA bound to soil (discussed in detail below). In our study, the number and concentration of amplified eDNA samples were statistically indistinguishable across wet and dry seasons. Researchers may consider using mixed-effects models to investigate effects of multiple factors on eDNA concentration [60].

## Historical eDNA

Historical eDNA leaching from sediment originating from past populations is a potential eDNA source. eDNA in soil may last for decades to centuries [62]. For eDNA persistence in water, studies have shown high variability; eDNA of the target species could not be detected within an hour upon removal of individuals [45], while other studies showed persistence for 24 hours [63], 48 hours [56, 64, 65], and even 45 days [66]. For soil, eDNA persistence highly depends on the nature and origin of the sediment [67]. In our study, eDNA was amplified from two HR streams with no confirmation of turtles being present. Based on our knowledge of these streams (turtles historically present) and survey results (only 1/12 positive water samples, no turtles found), the persistence of historical DNA is a possible explanation for these results. Our result shows that false positives should be considered when interpreting eDNA data, especially in scenarios with sporadic detection.

## False negatives

False negatives are a concern when interpreting results of eDNA studies, especially for low-density taxa. The volume and number of replicates of water sampled have been pointed out to be important factors affecting detection rate of target species [68–71]. A larger sampling volume (45–1000 L) is expected to boost eDNA detection rate [68, 72–74]. However, there is a trade-off between water volume and factors such as filtration efficiency and risk of PCR inhibition. The use of relatively small sampling volumes is common in eDNA studies, with sampling volumes of <1 L [75, 76] to 1–2 L [13, 31–33, 77]. Mächler et al. [69] suggested the optimal sampling volume depends on the species-habitat combination, therefore, pilot studies are required for each study system. Studies comparing sample volume and detection rate found number of species detected saturated at 68 L in tropical streams and rivers [68] and 1 L in natural, small rivers [71].

The sampling environment in our study (small, narrow streams) is similar to that of Sakata et al. [71]. The consistent detection of *P. megacephalum* eDNA in our study suggests that 1 L of water is sufficient to give present/absence data in narrow streams. However, we cannot rule out the possibility of false negatives due to insufficient water volume. We emphasize the importance of conducting pilot field experiments for each study system to determine the minimum volume of water sample required to detect a population of target species.

## Limitations of the study

eDNA quantification in qPCR can be estimated by building a standard curve from samples with known concentration. Optimal qPCR efficiency (90–110%) and $R^2$ (> 0.990) are two indicators of qPCR with robust results [39]. Despite using the same protocol, reagents and

equipment of a previously developed assay (average efficiency: 94.15%, $R^2$: 0.97; [29]), four amplified eDNA samples in this study had lower values (efficiency: 82.8–87.6%). This suggests eDNA quantification was affected in some samples. We were unable to pinpoint the cause of these results. Future researchers should be aware and ensure high efficiency and $R^2$ when performing qPCR to increase reliability of data. Adjustment of laboratory protocol (primer design, reaction volume and reagent concentration) may be required when amplifying environmental water samples.

## Conclusion

In this study, we field tested and validated a species-specific eDNA assay to detect and monitor the endangered *P. megacephalum*: sites with known turtle populations had positive eDNA detection, while new turtle populations were discovered in some streams. These results will be used to guide conservation of *P. megacephalum*; in Hong Kong, these data will be shared with the relevant governmental departments and NGOs to direct conservation efforts to sites with populations, while in other countries within the range of *P. megacephalum*, we will explore extending the use of this eDNA assay to locate wild populations.

We also highlight the potential difficulties interpreting eDNA results from natural environments. Even in streams with known turtle populations, eDNA detection can be inconsistent, raising the importance of understanding the study system—ecology of the organism, environmental conditions of the collection sites, and their influence on eDNA detection. As environmental factors and their interactions vary among sites, we believe each study site is unique in terms of eDNA distribution, degradation, production, transportation, and dilution. To faithfully interpret eDNA results, one must understand eDNA behaviour spatially and temporary in the natural environment. We caution against applying generalizations of eDNA properties across sites, and recommend performing pilot studies to design a site-specific eDNA sampling and data analysis strategy. eDNA methods are powerful tools that have tremendous potential to aid in species detection, monitoring, and conservation. However, like many scientific approaches, using eDNA methods blindly or without appropriate preliminary testing will result in erroneous results and interpretation. We believe that with advances in and refinement of methods, eDNA-based surveys will make positive contributions to the conservation of turtles and other organisms.

## Supporting information

**S1 Table. A validation checklist for qPCR assay.**
(XLSX)

**S2 Table. qPCR quantification of *Platysternon megacephalum ND4* assay in eDNA survey.**
(XLSX)

**S3 Table. Capture rate of *Platysternon megacephalum* trapping surveys.**
(XLSX)

## Acknowledgments

We thank the following people for invaluable discussions on eDNA: Kathryn Stewart, Caren Goldberg, David Pilliod, Tracie Seimon, and Brian Horne. We would also like to thank the team that helped with fieldwork: Jose Wellington Alves dos Santos, Anchalee Aowphol, Liu Lin, Li Ding, Fengyue Shu, Itzue Caviedes Solis, Abby Bauer, Hing Fong Cheung, Siu Chung Lo, and Siu Yin Yu.

## Author Contributions

**Conceptualization:** Yik-Hei Sung, Jonathan J. Fong.

**Formal analysis:** Ivan P. Y. Lam.

**Funding acquisition:** Yik-Hei Sung, Jonathan J. Fong.

**Investigation:** Ivan P. Y. Lam, Yik-Hei Sung.

**Methodology:** Ivan P. Y. Lam, Jonathan J. Fong.

**Resources:** Jonathan J. Fong.

**Supervision:** Jonathan J. Fong.

**Writing – original draft:** Ivan P. Y. Lam.

**Writing – review & editing:** Yik-Hei Sung, Jonathan J. Fong.

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
