## [Decision Letter · Decision Letter 0]

8 Jul 2021

PONE-D-21-13336

Using eDNA techniques to find the endangered big-headed turtles (Platysternon megacephalum)

PLOS ONE

Dear Dr. FONG,

Before rendering my decision, I apologize to the authors for the unreasonably slow turn-around time for this decision. In the first half of 2021, and continued life in a global pandemic, I have endured an unrelenting string of personal and professional crises that unfortunately diverted my attention from this manuscript. And while I had considered requesting the manuscript shift to another editor, that would've only delayed things further. I apologize to the authors for extreme delay in rendering a decision. Nevertheless, three reviewers were solicited to adjudicate this manuscript, and all three accepted. However one was unable to complete their review in time. Two reviewers provided comprehensive, instructive reviews and should be applauded for their efforts. These reviews are largely congruent with one another, and having read the manuscript I fully agree with their assessments.

Both reviewers note that this manuscript is an important contribution, as it focuses on a woefully understudied species that is of clear conservation concern. Given its status, low population numbers, and difficulty to detect using conventional means, environmental DNA analysis would seem to be an important advancement for monitoring this species. However, for eDNA approaches to be successful, the methodology must be fully-validated and reliable. Both reviewers (and I concur) that 1) critical information required (e.g. LoD and LoQ) to fully assess the assay's performance are lacking, and 2) other parameters (e.g. efficiency and R2) may indicate an assay that is not fully optimized. The reviewers both provide critical recommendations around these issues (in addition to other issues, both trivial and non-trivial) to increase transparency and to allow for a full vetting of the assay itself. Moreover, other critical issues regarding study design (i.e. sample sizes and water volumes) and interpretation (i.e. the treatment of "false negatives") require adjudication and, at the very least, contextualization in the discussion before this can be fully considered for publication in this volume.

Given the above, both reviewers argue for Major Revisions, and I concur. Therefore, we invite you to submit a revised version of the manuscript that addresses the points raised during the review process.

We look forward to receiving your revised manuscript.

Kind regards,

Mark A. Davis, Ph.D.

Academic Editor

PLOS ONE

Journal Requirements:

Reviewers' comments:

Reviewer's Responses to Questions

**Comments to the Author**

1. Is the manuscript technically sound, and do the data support the conclusions?

Reviewer #1: Partly

Reviewer #2: Partly

2. Has the statistical analysis been performed appropriately and rigorously? 

Reviewer #1: Yes

Reviewer #2: I Don't Know

3. Have the authors made all data underlying the findings in their manuscript fully available?

Reviewer #1: Yes

Reviewer #2: Yes

4. Is the manuscript presented in an intelligible fashion and written in standard English?

Reviewer #1: Yes

Reviewer #2: Yes

5. Review Comments to the Author

Reviewer #1: General comments

I have reviewed the manuscript ‘Using eDNA techniques to find the endangered big-headed turtles (Platysternon megacephalum)’. This is an interesting manuscript that uses environmental DNA (eDNA) analysis to assess the distribution of the endangered big-headed turtle and evaluate whether eDNA analysis can be used to assess relative abundance. The study indicates that big-headed turtle can be detected using eDNA analysis, eDNA results can potentially reveal new populations as well as existing populations, and eDNA analysis shows promise for relative abundance estimation. eDNA detection was inconsistent and the authors presented several suggestions for this. However, I have concerns about the experimental design, sampling strategy and qPCR analysis that may be responsible for inconsistent and negative detections.

First and foremost, the authors only took one 1 L water sample from each stream. This is extremely low sampling effort for lotic systems with no biological replication. Without a pilot field experiment, mesocosm experiment or power analysis, how can the authors be certain that this was enough water to detect smaller populations of big-headed turtle? I appreciate the authors probably cannot repeat the study due to cost, time and effort that was required to collect and process samples. However, they should at least discuss the implications of low sampling effort.

Second, the limit of detection and method used to establish this are not clearly reported in the manuscript. It is unclear whether negative samples simply fell below the limit of detection but yielded a Ct value or did not yield a Ct value at all. If the limit of detection was relatively high and authors classed samples falling below this as negative, then this indicates the assay should have been redesigned or optimised to increase sensitivity. The lowest standard curve point should really be 10 copies/μl or 1 copy/μl.

The range and average of R-squared and efficiency values for qPCR should also be reported in the main text. Having looked at the Supporting Information, both values were lower than what is acceptable for qPCR according to the MIQE guidelines (Bustin et al. 2009 Clinical Chemistry https://doi.org/10.1373/clinchem.2008.112797) with R-squared less than 0.990 and efficiency less than 90%. Low qPCR efficiency indicates poor primer design and non-optimal reagent concentration or reaction conditions whereas low R-squared indicates pipetting inaccuracy. Both of these will influence qPCR detection and quantification and should be discussed.

Overall, the authors seem to be too quick to assume that sporadic eDNA detections are false positives due to eDNA transport or ancient DNA from soil. More detections may have been observed from these sites with increased sampling effort or optimisation of the qPCR assay. Alternatively, contamination may have still occurred even though negative controls did not exhibit amplification. Without knowing how many negative controls were included, contamination cannot be ruled out. Furthermore, the authors have not stated how their gBlock was prepared and handled. Was this prepared and added to qPCR plates/tubes in a separate laboratory to eDNA samples?

The quality and resolution of all figures needs to be improved substantially before publication, and the authors should use past tense throughout the manuscript. There is currently a mix of past and present tense. I have a number of minor comments to help improve the clarity of the manuscript, which I have detailed in the specific comments to the authors below.

Specific comments

Abstract: The abstract would benefit from a sentence or two on the conservation status of the target species and the benefits of environmental DNA analysis for monitoring this species.

Line 19: Insert ‘analysis’ after ‘(eDNA)’.

Line 23: Change ‘are’ to ‘were’.

Line 24: Change ‘population’ to ‘populations’.

Line 42: Change ‘eDNA based’ to ‘eDNA-based’.

Line 71: Change ‘collecting’ to ‘survey’.

Line 79: Remove ‘column’.

Line 82: Change ‘0.45m’ to ‘0.45μm’.

Lines 82-87: How were filters removed from the filter funnels and divided into two pieces? Were gloved hands, forceps and/or scissors used?

Lines 93-94: For samples where both halves of the filter were extracted, were the extracts combined for qPCR or amplified independently?

Line 94: How many of each type of negative control were included?

Line 98: Change ‘in’ to ‘on’.

Lines 106-107: Write values as numerals, i.e. 1,000,000, 100,000, 10,000.

Line 149: Use ‘consistent’ instead of ‘continuous’?

Line 158: Change ‘streams.’ to ‘streams’.

Line 172: Did negative samples include those that amplified but fell below the limit of detection?

Line 173: What was the limit of detection?

Line 317: Change ‘spotty’ to ‘sporadic’.

Line 336: Change ‘result’ to ‘results’.

Line 341: Change ‘advances and refinement’ to ‘advances in and refinement of’.

Reviewer #2: Overall:

In the study, “Using eDNA techniques to find the endangered big-headed turtles (Platysternon megacephalum)” the authors used environmental DNA to locate endangered turtles. Overall the paper covers a taxon that has been relatively understudied using eDNA (I was very excited to see a turtle eDNA paper). The authors should be commended on a well-written paper that used eDNA over such a long-term study.

Below I provide more specific comments and request further clarification on laboratory procedures (field blanks, LOD) and statistical analysis.

Abstract:

L19: Could clarify “for crosschecking eDNA results”.

L19: Revise to add “and” to sentence, to read, “…were first identified with eDNA and then verified…”

Introduction:

L47-48: Revise ending of sentence to read, “..and potential use in population genetics…”

Methods:

L79: Do the authors mean that water samples were collected by submerging a whirl-pak bag to collect water (not surface water samples)? Then consider revising to something like, “One liter of water was collected from the water column…”

L92: What does “recovered a positive value” mean? That eDNA samples had to have concentrations greater than 0.00 ng/µL?

L94-95 or L78-79: Were field blanks collected?

L100-103: What was the volume of the eDNA added to each qPCR reaction?

Also, curious as to why 5 µL of TaqMan Environmental Master Mix was used? That seemed low compared to other eDNA papers.

L136: Were eDNA samples collected separately from trapping?

L146-147-Why not consider using occupancy modeling to analyze the data? There are multiple replications over years at multiple sites, some with detections and some with no detections. The authors could then assess detection and occupancy probability and use AIC or some other information criterion to assess the impacts of multiple a priori variables (pH, temperature, etc.) to assess model fit instead of correlation analyses?

A helpful reference might be: Akre et al. 2019 Concurrent visual encounter sampling validates eDNA selectivity and sensitivity for the endangered wood turtle (Glyptemys insculpta). PLoS ONE 14:e0215586. https://doi.org/10.1371/journal.pone.0215586

L148-150: What do the correlations look like when all the data are included vs. only those from KP streams? I can see from Figure 2 that detections in the other streams are limited, but it seems almost like cherry-picking data to only analyze some of the sampled streams.

Results:

L103-104: Were any of the samples inhibited? This is not mentioned again in the results.

L167 and section on qPCR: Please provide R2 and estimates of efficiency here. Such information is available in the supplemental information

L173: “all amplified water samples were above the limit of detection as validated in [30].” Could the authors please clarify the limit of detection? I looked at the reference and could not easily identify the LOD/LOQ, was it 10 copies/µL?

L189: Suggest editing sentence to read, “No amplification was detected in any of the 12 eDNA…”

Discussion:

This section needs some work. While there is good discussion of factors that are important for eDNA degradation, production, and transport, there should be further explanation of the results so that future research might gain new understanding and discussion of the shortcomings.

For example, only one water sample was collected at a time. How might the lack of replication influence the results? There are several papers that show that increasing volume of water and/or number of replicates increases detection.

How might the definition of “amplified” vs. “uncertain” from Figure 1 influence the results. Other eDNA researchers use similar classifications, but might the exclusion of the uncertain simples indicate low-abundance individuals producing little eDNA? What do the correlational analyses look like if these are included?

From the Supplemental Table 1. Some of the r2 and %efficiencies are lower than would be advised to attempt quantification. Generally, R2>0.99 and %efficiency should be 90-110% to attempt to quantify eDNA. How might this influence the results?

L244: replace “u” with “µ”

L245:Were any of the positive eDNA samples sequenced to confirm species? See the comment below, but I was surprised to read about false negative and false positives. As it seemed the assay worked well and the eDNA generally was confirmed by trapping/active searches.

L255-257: It may also be other factors like turtle biomass or age that influences this relationship.

L264-265: Others have found that temperature is an important reproductive cue and eDNA rates are higher on warmer temperatures. Not just that temperature causes degradation.

L271-273: Such laboratory experiments examining eDNA degradation and persistence have already been conducted; what future experiments should be done? Further explanation needed.

L286-288: Not sure this is true and requires further explanation on what environmental factors-because pH and temperature as measured in this study do not appear to influence eDNA. As others have found increased eDNA detection for low-abundance species during breeding seasons. As an example, see de Souza et al. 2016. Environmental DNA (eDNA) detection probability is influenced by seasonal activity of organisms. PLoS ONE 11:e0165273. https://doi.org/10.1371/journal.pone.0165273

L293 and section on eDNA transport: What about dilution? Were there noticeable differences between the wet and dry season? If few organisms are present, that might suggest that less eDNA is available in water and thus, during high flows it would be more difficult to detect eDNA (dilution).

L307-318: I think this section could be deleted or condensed and combined with a different section. It is not possible to assess whether deposited eDNA was resuspended and caused false positives.

L334-335: Was there evidence of false negatives and false positives? This came as a surprise to me as a reader.

Literature Cited:

Scientific names should be italicized in the references, see #11, 24, 47.

Figures:

Figure 1 and 2 were blurry and difficult to read. Not sure if this was a conversion issue within the submission stage.

I would have liked to see the correlation figures and not the raw data.

6. PLOS authors have the option to publish the peer review history of their article (what does this mean?). If published, this will include your full peer review and any attached files.

Reviewer #1: No

Reviewer #2: No

---

## [Author Response · Author response to Decision Letter 0]

20 Aug 2021

Reviewer #1: General comments

I have reviewed the manuscript ‘Using eDNA techniques to find the endangered big-headed turtles (Platysternon megacephalum)’. This is an interesting manuscript that uses environmental DNA (eDNA) analysis to assess the distribution of the endangered big-headed turtle and evaluate whether eDNA analysis can be used to assess relative abundance. The study indicates that big-headed turtle can be detected using eDNA analysis, eDNA results can potentially reveal new populations as well as existing populations, and eDNA analysis shows promise for relative abundance estimation. eDNA detection was inconsistent and the authors presented several suggestions for this. However, I have concerns about the experimental design, sampling strategy and qPCR analysis that may be responsible for inconsistent and negative detections.

First and foremost, the authors only took one 1 L water sample from each stream. This is extremely low sampling effort for lotic systems with no biological replication. Without a pilot field experiment, mesocosm experiment or power analysis, how can the authors be certain that this was enough water to detect smaller populations of big-headed turtle? I appreciate the authors probably cannot repeat the study due to cost, time and effort that was required to collect and process samples. However, they should at least discuss the implications of low sampling effort.

RESPONSE:

We agree the 1 L sampling effort is relatively low compared to currently published eDNA studies. This is one of the difficulties with a multi-year study; we had designed out study based on eDNA studies and guidelines 4 years ago, when many of these factors were not as well understood. We appreciate the reviewer’s understanding on the concern of cost, time and effort to repeat the experiment. To address the issue of low sampling effort, we conducted a literature review on sampling volume and eDNA detection efficiency. We found that using large-volume sampling strategy, for instances, 45 L to 1000 L, can enhance eDNA detection. However, with larger volumes, filtration efficiency will decrease and risk of PCR inhibition will increase. We found two studies (Cantera et al. 2019; Sakata et al. 2020) comparing the sampling volume and eDNA detection rate. The result of these two studies vary greatly in terms of the volume when number of species detection is saturated (1L in Sakata et al. 2020, versus 68L in Cantera et al. 2019).

In our study, the system is similar to that in Sakata et al. (2020), both being small, narrow streams. In our study, we included sites with stable Platysternon megacephalum populations, and 1 L of water gave us consistent detection of turtle eDNA, providing some confidence that in those streams 1 L is sufficient. Nevertheless, the sporadic eDNA detection at other sites may be associated with insufficient water sampling, and we hedge our interpretation by including the possibility of false negatives. We organized the discussion above and included a new paragraph “False negative” in the Discussion section.

Reference:

Cantera I, Cilleros K, Valentini A, Cerdan A, Dejean T, Iribar A, et al. (2019). Optimizing environmental DNA sampling effort for fish inventories in tropical streams and rivers. Scientific Reports 9:3085. doi: 10.1038/s41598-019-39399-5

Sakata M, Watanabe T, Maki N, Ikeda K, Kosuge T, Okada H, et al. (2020). Determining an effective sampling method for eDNA metabarcoding: a case study for fish biodiversity monitoring in a small, natural river. Limnology 22(9). doi: 10.1007/s10201-020-00645-9

Second, the limit of detection and method used to establish this are not clearly reported in the manuscript. It is unclear whether negative samples simply fell below the limit of detection but yielded a Ct value or did not yield a Ct value at all. If the limit of detection was relatively high and authors classed samples falling below this as negative, then this indicates the assay should have been redesigned or optimised to increase sensitivity. The lowest standard curve point should really be 10 copies/μl or 1 copy/μl.

RESPONSE:

We adopted a previously developed assay, and the limit of detection was tested in Lam et al. (2020). The testing process are summarized as follows: A synthetic gene fragment containing the primer and probe binding region of target species P. megacephalum was made to test for analytical sensitivity. The assay was then tested against a 4-level dilution series (10,000, 1000, 100, and 10 copies per μL) of the synthetic gene with 20 replicates for each concentration level. Lam et al. (2020) found that the analytical sensitivity of the assay at 10 copies / μL was 0.95, indicating a reliable detection of eDNA down to 10 copies / μL. In this study, we estimated eDNA quantity using a standard curve. To reduce the variation of standard curve, we used 100 copies per μL as the lowest level to ensure 100% amplification among replicates.

In processing qPCR data, all samples classified as negative had no fluorescence signal (no Ct value). In other words, we observed no samples with qPCR fluorescence signal that fell below threshold value calculated by the qPCR software. We clarified these details in the methodology and results section.

The range and average of R-squared and efficiency values for qPCR should also be reported in the main text. Having looked at the Supporting Information, both values were lower than what is acceptable for qPCR according to the MIQE guidelines (Bustin et al. 2009 Clinical Chemistry https://doi.org/10.1373/clinchem.2008.112797) with R-squared less than 0.990 and efficiency less than 90%. Low qPCR efficiency indicates poor primer design and non-optimal reagent concentration or reaction conditions whereas low R-squared indicates pipetting inaccuracy. Both of these will influence qPCR detection and quantification and should be discussed.

RESPONSE:

In this study, R-squared (0.89–0.99) and efficiency (82.8%–109%) values were relatively low compared to the assay developed in Lam et al. (2020) (R-squared: 0.97; efficiency: 94.15%). Although we followed the same laboratory protocol as Lam et al. (2020), the qPCR efficiency had high variation and R-square was low. We are unable to redo this labwork, so we follow the recommendation from Review #1, to discuss how low R-squared and efficiency values will influence qPCR detection and quantification.

To minimize the impact of low qPCR efficiency, we analyze a subset of the data excluding these low qPCR efficiency samples, only including samples with 90–100% efficiency. Four amplified eDNA samples were removed in the new dataset. The results were consistent, only with differences in rs and p-value. The updated result are as follows:

1. A significant, positive correlation (rs = 0.718, p = 0.02) between the number of samples with amplified eDNA and mean capture rate.

2. Environmental variables (temperature, and pH) have no significant relationship with eDNA quantity datasets (KP5, and all KP streams).

3. No significant relationship between eDNA quantity and capture rate for the dataset KP5 (rs = -0.036, p = 0.90)

4. A significant, positive relationship between eDNA quantity and capture rate for dataset in all KP streams (rs = 0.265, p = 0.046).

We included the discussion of non-optimal qPCR R-square, efficiency in a new paragraph “Limitations of the study” in discussion section; and updated the result of statistical analysis in the result section.

Overall, the authors seem to be too quick to assume that sporadic eDNA detections are false positives due to eDNA transport or ancient DNA from soil. More detections may have been observed from these sites with increased sampling effort or optimisation of the qPCR assay.

RESPONSE:

We agree more detections may have been observed with a larger sampling volume, and further adjustment in qPCR protocol. We included the potential influence of sampling volume, and implication of non-optimal qPCR result in the discussion section.

Alternatively, contamination may have still occurred even though negative controls did not exhibit amplification. Without knowing how many negative controls were included, contamination cannot be ruled out.

RESPONSE:

For every round of DNA extraction, we included one negative control alongside field-collected samples. The DNA concentration of negative control, together with environmental samples were measured by Qubit 3.0. No DNA was detected in negative controls.

In every qPCR event, we included 2–3 negative controls per 96-well plate. Fluorescent signal was not detected in any negative controls, and no Ct value was calculated based on qPCR software.

We clarified the information above in the methodology section.

Furthermore, the authors have not stated how their gBlock was prepared and handled. Was this prepared and added to qPCR plates/tubes in a separate laboratory to eDNA samples?

RESPONSE:

A synthetic gene fragment containing primer and probe binding site was made by a local company (Tech Dragon Limited; Hong Kong) for producing the standard curve. Serial dilution of the synthetic gene was conducted prior to qPCR event a week before.

The dilution of synthetic gene was added to qPCR 96-well plates in the same laboratory with eDNA samples. Nevertheless, synthetic gene and environmental samples were added, and stored separately to minimize contamination risk, with regular cleaning of lab benches and equipment with a 10% bleach solution.

We added the details above in the methodology section. 

The quality and resolution of all figures needs to be improved substantially before publication, and the authors should use past tense throughout the manuscript. There is currently a mix of past and present tense. I have a number of minor comments to help improve the clarity of the manuscript, which I have detailed in the specific comments to the authors below.

RESPONSE:

We believe the resolution of figures is a conversion problem during submission. We submitted a better quality of figures in re-submission. We apologies for using inconsistent tenses throughout the manuscript. We screened grammatical mistakes in our amended manuscript.

Specific comments

Abstract: The abstract would benefit from a sentence or two on the conservation status of the target species and the benefits of environmental DNA analysis for monitoring this species.

RESPONSE:

We add text to the abstract as suggested. “The results from this study are important for identifying priority sites for protection and further research in Hong Kong.”

Line 19: Insert ‘analysis’ after ‘(eDNA)’.

RESPONSE:

Revised as suggested.

Line 23: Change ‘are’ to ‘were’.

RESPONSE:

Revised as suggested.

Line 24: Change ‘population’ to ‘populations’.

RESPONSE:

Revised as suggested.

Line 42: Change ‘eDNA based’ to ‘eDNA-based’.

RESPONSE:

Revised as suggested.

Line 71: Change ‘collecting’ to ‘survey’.

RESPONSE:

Revised as suggested.

Line 79: Remove ‘column’.

RESPONSE:

Revised as suggested.

Line 82: Change ‘0.45m’ to ‘0.45μm’.

RESPONSE:

Revised as suggested.

Lines 82-87: How were filters removed from the filter funnels and divided into two pieces? Were gloved hands, forceps and/or scissors used?

RESPONSE:

Filters were removed from the funnel with sterile forceps. Each filter was then folded and stored in 95% ethanol. Prior to DNA extraction, the filter paper was divided into two equal parts with sterile scissors and forceps. Gloved hands were used in all laboratory procedures. We clarified the details above in the methodology section.

Lines 93-94: For samples where both halves of the filter were extracted, were the extracts combined for qPCR or amplified independently?

RESPONSE:

We followed previous protocols that cut the filter in half, so there are two chances to extract DNA. In this study, there were no failure of DNA extraction. Hence, all the extraction used in this study are originated from half of the filter only. We clarified the details above in the methodology section.

Line 94: How many of each type of negative control were included?

RESPONSE:

In each DNA extraction event, we at most extract 23 eDNA samples, plus one negative control. We performed DNA quantification using Qubit 3.0 after each DNA extraction event, and include one negative control each time. For each qPCR event (a 96-well plate), we include 2-3 negative controls. We clarified the details above in the methodology section.

Line 98: Change ‘in’ to ‘on’.

RESPONSE:

Revised as suggested.

Lines 106-107: Write values as numerals, i.e. 1,000,000, 100,000, 10,000.

RESPONSE:

Revised as suggested.

Line 149: Use ‘consistent’ instead of ‘continuous’?

RESPONSE:

Revised as suggested.

Line 158: Change ‘streams.’ to ‘streams’.

RESPONSE:

Revised as suggested.

Line 172: Did negative samples include those that amplified but fell below the limit of detection?

RESPONSE:

Samples that we classified as “not amplified” could either have zero Ct value or have a Ct value that is below the threshold. All not amplified samples in our study had a zero Ct value.

Line 173: What was the limit of detection?

RESPONSE:

The limit of detection of the assay was tested against a 4-level dilution series (10,000, 1000, 100, and 10 copies per μL) of the synthetic gene with 20 replicates for each concentration level (Lam et al. 2020). The analytical sensitivity of at 10 copies/μL = 0.95, indicating a reliable detection to 10 copies/μL (Lam et al. 2020).

Line 317: Change ‘spotty’ to ‘sporadic’.

RESPONSE:

Revised as suggested.

Line 336: Change ‘result’ to ‘results’.

RESPONSE:

Revised as suggested.

Line 341: Change ‘advances and refinement’ to ‘advances in and refinement of’.

RESPONSE:

Revised as suggested.

 

Reviewer #2: Overall:

In the study, “Using eDNA techniques to find the endangered big-headed turtles (Platysternon megacephalum)” the authors used environmental DNA to locate endangered turtles. Overall the paper covers a taxon that has been relatively understudied using eDNA (I was very excited to see a turtle eDNA paper). The authors should be commended on a well-written paper that used eDNA over such a long-term study.

Below I provide more specific comments and request further clarification on laboratory procedures (field blanks, LOD) and statistical analysis.

Abstract:

L19: Could clarify “for crosschecking eDNA results”.

RESPONSE:

Apart from the presence or absence of eDNA, we performed field survey (turtle trapping or active searching) at each study site. The term “crosschecking” here means using result of field survey to verify the presence or absence of turtle in each study site. We removed the sentence to avoid confusion, and clarified the details in the methodology section.

L19: Revise to add “and” to sentence, to read, “…were first identified with eDNA and then verified…”

RESPONSE:

Revised as suggested.

Introduction:

L47-48: Revise ending of sentence to read, “..and potential use in population genetics…”

RESPONSE:

Revised as suggested.

Methods:

L79: Do the authors mean that water samples were collected by submerging a whirl-pak bag to collect water (not surface water samples)? Then consider revising to something like, “One liter of water was collected from the water column…”

RESPONSE:

We amended the sentence as suggested. The sentence is revised as “One liter of water was collected from the water column (Davy et al. 2015; Laramie et al. 2015; Feng et al. 2019; Kirtane et al. 2019), using a sterile 50-mL conical tube”

L92: What does “recovered a positive value” mean? That eDNA samples had to have concentrations greater than 0.00 ng/µL?

RESPONSE:

After DNA extraction, we quantify concentration of eDNA samples using Qubit 3.0 Fluorometer with dsDNA HS Assay Kit. The limit of detection of the kit ranged from 0.2 to 100ng. Hence, the term “recovered a positive value” means eDNA samples contain at least 0.2ng/µL eDNA.

We added the detection limit of Qubit in the sentence. The sentence is revised as “Samples were used in subsequent quantitative PCR (qPCR) analyses if DNA quantification recovered a positive value greater than the detection limit of fluorometer (>0.2ng).”

L94-95 or L78-79: Were field blanks collected?

RESPONSE:

We agree that including field blanks in our study would be much better, but we unfortunately did not do this. In our study, each collecting trip was from a single site, and all samples from a single collecting trip was filtered at the same time. In this case, the use of a field blank would help us detect contamination between samples from a single site. The analysis of our data was grouped site-by-site. Therefore, although it would have been more methodologically sound to collect a field blank, we believe that not including them did not dramatically affect our results.

L100-103: What was the volume of the eDNA added to each qPCR reaction?

RESPONSE:

In each qPCR reaction, 1 µL of eDNA extraction was added. 

We included the volume of eDNA used in the sentence. The sentence is revised as “Each qPCR reaction included 1 μL eDNA template, 5 μL TaqMan® Environmental Master Mix (Thermo Fisher Scientific Inc; Waltham, MA, USA), 900 nM of each primer, 250 nM of probe, and enough autoclaved Milli-Q® water to make a final volume of 10 μL.”

Also, curious as to why 5 µL of TaqMan Environmental Master Mix was used? That seemed low compared to other eDNA papers.

RESPONSE:

We adopted a previously developed qPCR assay in Lam et al. (2020), and followed the qPCR protocol as suggested. In this paper, 5 µL was tested and considered to be sufficient to produce consistent results.

L136: Were eDNA samples collected separately from trapping?

RESPONSE:

Yes. In this study, eDNA samples were collected separately from turtle trapping. This was because we utilized some data from a long-term population monitoring programme. We clarified this in the text.

L146-147-Why not consider using occupancy modeling to analyze the data? There are multiple replications over years at multiple sites, some with detections and some with no detections. The authors could then assess detection and occupancy probability and use AIC or some other information criterion to assess the impacts of multiple a priori variables (pH, temperature, etc.) to assess model fit instead of correlation analyses?

A helpful reference might be: Akre et al. 2019 Concurrent visual encounter sampling validates eDNA selectivity and sensitivity for the endangered wood turtle (Glyptemys insculpta). PLoS ONE 14:e0215586. https://doi.org/10.1371/journal.pone.0215586

RESPONSE:

As suggested, we conducted a dynamic occupancy modeling using R package “unmarked”. We determined factors affecting detection probability using pH, temperature, and season (wet and dry) as covariate, while assuming constant for colonization and extinction probability in the model. Both models created using pH and temperature as covariate were regarded as the best model (lowest AICc, with difference <2). However, both of the covariates did not significantly affect detection probability (p>0.05). We included the information above in the methodology and result section.

L148-150: What do the correlations look like when all the data are included vs. only those from KP streams? I can see from Figure 2 that detections in the other streams are limited, but it seems almost like cherry-picking data to only analyze some of the sampled streams.

RESPONSE:

In this study, correlation analysis was tested against combination of multiple variables. We believed that the most appropriate comparison was sites that we had estimates of the population size. The variable “capture rate”, was obtained from a long-term population monitoring programme, which were only conducted in KP streams, but not HR and UN streams. Therefore, it is not possible to compare eDNA quantity of the entire dataset with capture rate (data missing in HR and UN stream).

We summarize all possible correlation analysis combination as follows:

1. Number of samples with amplified eDNA, versus mean capture rate

- Only KP stream dataset were useful.

- A significant, positive relationship was found.

2. Environmental variables (temperature, and pH), versus eDNA quantity

- Both the entire dataset, and KP stream dataset were useful

- No significant relationship was found for both dataset

3. eDNA quantity, versus capture rate

- KP stream dataset, and KP5 stream dataset were useful.

- A significant, positive relationship was found in KP stream dataset.

- No significant relationship was found in KP5 stream dataset.

We organized and made our text clearer in the manuscript to reflect this.

Results:

L103-104: Were any of the samples inhibited? This is not mentioned again in the results.

RESPONSE:

We included the TaqMan® Exogenous Internal Positive Control (IPC) to test for inhibition in each qPCR reaction. As no IPC showed inhibition in this study, no eDNA samples were regarded as inhibited. We included a sentence “No eDNA samples exhibited inhibition.” In the “Results” section. 

L167 and section on qPCR: Please provide R2 and estimates of efficiency here. Such information is available in the supplemental information

RESPONSE:

We added the required details (qPCR efficiency: 96.99% ± 5.19; R2: 0.957 ± 0.03) in the methodology section.

L173: “all amplified water samples were above the limit of detection as validated in [30].” Could the authors please clarify the limit of detection? I looked at the reference and could not easily identify the LOD/LOQ, was it 10 copies/µL?

RESPONSE:

Yes. According to Lam et al. (2020), the assay was then tested against a 4-level dilution series (10,000, 1000, 100, and 10 copies per μL) of the synthetic gene with 20 replicates for each concentration level. The analytical sensitivity of the assay at 10 copies / µL was 0.95.

We included the details of limit of detection in the sentence. The sentence is revised as “All amplified water samples were above the limit of detection (10 copies / μL) as validated in [30], while all negative water samples yielded no qPCR fluorescence signal.”

L189: Suggest editing sentence to read, “No amplification was detected in any of the 12 eDNA…”

RESPONSE:

Revised as suggested.

Discussion:

This section needs some work. While there is good discussion of factors that are important for eDNA degradation, production, and transport, there should be further explanation of the results so that future research might gain new understanding and discussion of the shortcomings.

For example, only one water sample was collected at a time. How might the lack of replication influence the results? There are several papers that show that increasing volume of water and/or number of replicates increases detection.

RESPONSE:

We agree the 1 L sampling effort and no replicates could have influenced our results. This is one of the difficulties with a multi-year study; we had designed out study based on eDNA studies and guidelines 4 years ago, when many of these factors were not as well understood. We appreciate the reviewer’s understanding on the concern of cost, time and effort to repeat the experiment. To address the issue of low sampling effort, we conducted a literature review on sampling volume and eDNA detection efficiency. We found that using large-volume sampling strategy, for instances, 45 L to 1000 L, can enhance eDNA detection. However, with larger volumes, filtration efficiency will decrease and risk of PCR inhibition will increase. We found two studies (Cantera et al. 2019; Sakata et al. 2020) comparing the sampling volume and eDNA detection rate. The result of these two studies vary greatly in terms of the volume when number of species detection is saturated (1L in Sakata et al. 2020, versus 68L in Cantera et al. 2019).

In our study, the system is similar to that in Sakata et al. (2020), both being small, narrow streams. In our study, we included sites with stable Platysternon megacephalum populations, and 1 L of water gave us consistent detection of turtle eDNA, providing some confidence that in those streams 1 L is sufficient. Nevertheless, the sporadic eDNA detection at other sites may be associated with insufficient water sampling, and we hedge our interpretation by including the possibility of false negatives. We organized the discussion above and included a new paragraph “False negative” in the Discussion section.

Reference:

Cantera I, Cilleros K, Valentini A, Cerdan A, Dejean T, Iribar A, et al. (2019). Optimizing environmental DNA sampling effort for fish inventories in tropical streams and rivers. Scientific Reports 9:3085. doi: 10.1038/s41598-019-39399-5

Sakata M, Watanabe T, Maki N, Ikeda K, Kosuge T, Okada H, et al. (2020). Determining an effective sampling method for eDNA metabarcoding: a case study for fish biodiversity monitoring in a small, natural river. Limnology 22(9). doi: 10.1007/s10201-020-00645-9

How might the definition of “amplified” vs. “uncertain” from Figure 1 influence the results. Other eDNA researchers use similar classifications, but might the exclusion of the uncertain simples indicate low-abundance individuals producing little eDNA? What do the correlational analyses look like if these are included?

RESPONSE:

We agree that uncertain samples may have come from sites with a low-abundance of individuals. As a result, we modify the analysis to include data from uncertain eDNA samples in correlation analysis, as shown in manuscript.

Here we provide a comparison of result between inclusion, and exclusion of uncertain samples in our correlation analysis: 

1. Number of samples with amplified eDNA, versus mean capture rate

- A significant, positive relationship was found in both inclusion, and exclusion of uncertain sample dataset.

2. Environmental variables (temperature, and pH), versus eDNA quantity

- No significant relationship was found in both inclusion, and exclusion of uncertain sample dataset.

3. eDNA quantity, versus capture rate

- A significant, positive relationship was found in KP in both inclusion, and exclusion of uncertain sample dataset.

- No significant relationship was found in KP5 stream of both inclusion, and exclusion of uncertain sample dataset.

In summary, the result of overall correlation analysis was the same between inclusion, and exclusion of uncertain sample dataset.

We clarified we have included data from uncertain eDNA samples in correlation analysis by adding the following sentences at the beginning of the paragraph: “We performed correlation analysis using dataset including eDNA quantification from uncertain samples, followed by dataset excluding uncertain samples. Both datasets shown the same result, statistics here were from dataset including uncertain samples.”

From the Supplemental Table 1. Some of the r2 and %efficiencies are lower than would be advised to attempt quantification. Generally, R2>0.99 and %efficiency should be 90-110% to attempt to quantify eDNA. How might this influence the results?

RESPONSE:

In this study, R-squared (0.89–0.99) and efficiency (82.8%–109%) values were relatively low compared to the assay developed in Lam et al. (2020) (R-squared: 0.97; efficiency: 94.15%). Although we followed the same laboratory protocol as Lam et al. (2020), the qPCR efficiency had high variation and R-square was low. We are unable to redo this labwork, so we follow the recommendation from Review #1, to discuss how low R-squared and efficiency values will influence qPCR detection and quantification.

To minimize the impact of low qPCR efficiency, we analyze a subset of the data excluding these low qPCR efficiency samples, only including samples with 90–100% efficiency. Four amplified eDNA samples were removed in the new dataset. The results were consistent, only with differences in rs and p-value. The updated result are as follows:

1. A significant, positive correlation (rs = 0.718, p = 0.02) between the number of samples with amplified eDNA and mean capture rate.

2. Environmental variables (temperature, and pH) have no significant relationship with eDNA quantity datasets (KP5, and all KP streams).

3. No significant relationship between eDNA quantity and capture rate for the dataset KP5 (rs = -0.036, p = 0.90)

4. A significant, positive relationship between eDNA quantity and capture rate for dataset in all KP streams (rs = 0.265, p = 0.046).

We included the discussion of non-optimal qPCR R-square, efficiency in a new paragraph “Limitations of the study” in discussion section; and updated the result of statistical analysis in the result section.

L244: replace “u” with “µ”

RESPONSE:

Revised as suggested.

L245:Were any of the positive eDNA samples sequenced to confirm species? See the comment below, but I was surprised to read about false negative and false positives. As it seemed the assay worked well and the eDNA generally was confirmed by trapping/active searches.

RESPONSE:

We agree that we would be more confident if we sequence amplified positive eDNA samples, we have not done so in this study.

The reason behind discussing false negative and false positive is to be more conservative. As we highlighted potential difficulties interpreting eDNA results, we caution against applying generalizations of eDNA properties across sites, and recommend performing pilot studies to design a site-specific eDNA sampling and data analysis strategy. We prefer leaving some room for other possibilities, rather than solely relying on eDNA result until a promising refinement of eDNA method was adopted in eDNA surveys.

L255-257: It may also be other factors like turtle biomass or age that influences this relationship.

RESPONSE:

We agree that other factors (turtle biomass, age, etc.) as suggested can influence the relationship between eDNA quantity and turtle capture rate. Upon literature review, Maruyama et al. (2014), Klymus et al. (2015), and Toshiaki et al. (2018) successfully demonstrated fish biomass and age class as two of factors affecting eDNA production. We believed the findings could be applied to turtles as well.

We discussed life stage as one of the factors in influencing eDNA production in the discussion – eDNA production section. We revised the sentence as “These results suggest that eDNA methods remain promising to aid in estimating population size or biomass, but need to be refined based on differences in the habitat and species ecology.”

L264-265: Others have found that temperature is an important reproductive cue and eDNA rates are higher on warmer temperatures. Not just that temperature causes degradation.

RESPONSE:

We agree temperature is also a factor increasing eDNA production rate. Robson et al. (2016) and Lacoursière‐Roussel et al. (2016) documented the findings. We discussed temperature as a factor of increasing eDNA production in the discussion - eDNA production section.

To clarify temperature can both cause eDNA degradation and increase production, we made a note on the sentence and it is revised as “Multiple studies indicated that higher water temperature increases eDNA degradation rate (see also eDNA Production below), while other studies did not find any significant relationship.)

L271-273: Such laboratory experiments examining eDNA degradation and persistence have already been conducted; what future experiments should be done? Further explanation needed.

RESPONSE:

Laboratory experiments investigating eDNA degradation and persistence were often conducted in ideal and controlled environment, which may not reflect realistic field conditions. As multiple factors (other than temperature, pH, dissolved oxygen, etc.) potentially affecting eDNA concentration (both increase and decrease), experiments focusing on a few factors may not be applicable in practice. We suggest in situ pilot experiment conducted in sites with known target species abundance. The pilot study will provide information on eDNA fluctuation (including both eDNA degradation and persistence) in that site. With the aid of field surveys with enough survey effort, relationship between eDNA concentration and species abundance can be established. 

We added the discussion above in the conclusion section, and deleted the sentence “To advance our knowledge of environmental factors on eDNA degradation, we suggest controlled laboratory experiments, followed by mesocosm-type experiments in the field.”

L286-288: Not sure this is true and requires further explanation on what environmental factors-because pH and temperature as measured in this study do not appear to influence eDNA. As others have found increased eDNA detection for low-abundance species during breeding seasons. As an example, see de Souza et al. 2016. Environmental DNA (eDNA) detection probability is influenced by seasonal activity of organisms. PLoS ONE 11:e0165273. https://doi.org/10.1371/journal.pone.0165273

RESPONSE:

As the target species has a relatively short breeding season (May to July). We believed our sampling effort (twice in wet season, and twice in dry season) was not frequent enough to detect changes during the breeding season. We added the literature suggested in the paragraph to provide more background information of eDNA production during breeding season.

L293 and section on eDNA transport: What about dilution? Were there noticeable differences between the wet and dry season? If few organisms are present, that might suggest that less eDNA is available in water and thus, during high flows it would be more difficult to detect eDNA (dilution).

RESPONSE:

We agreed that eDNA may be more difficult to detect during high flows in wet season, especially for low-abundance species. If we compare the number of amplified eDNA samples between wet and dry season, there were no noticeable difference in this study (19 and 18 amplified samples respectively). On the other hand, if we compare the eDNA concentration, wet season generally yield a higher concentration. We caution interpreting the comparison result in the effect of dilution as it is outside the scope of this study. Nevertheless, we would like to discuss the effect of dilution on eDNA concentration, as it may affect eDNA detection probability.

eDNA is expected to be diluted under faster water flow, hence reducing eDNA detection probability, or even causing false negatives. While at the same time, an opposite outcome caused by fast water flow may be re-suspending precipitated eDNA or buried eDNA from soil, hence increasing eDNA detection probability. False positive may be possible if historical eDNA were amplified when target species are no longer there anymore. As the effect of dilution may cause complicated eDNA fluctuation in natural environment, we suggest researchers using mixed-effect models to address multiple factors affecting eDNA concentration at the same time, as shown in Curtis et al. 2020.

We organized the discussion above and include a new paragraph in the discussion – eDNA Dilution.

L307-318: I think this section could be deleted or condensed and combined with a different section. It is not possible to assess whether deposited eDNA was resuspended and caused false positives.

RESPONSE:

We agree it is not possible to assess whether deposited eDNA was resuspended, and caused false positives. The reason behind of discussing historical eDNA is to raise the possibility based on our eDNA and field survey data (sporadic eDNA detection, no turtles found in subsequent field survey). Although we cannot definitively prove this is the reason, we think it is useful to raise this possibility to the reader.

L334-335: Was there evidence of false negatives and false positives? This came as a surprise to me as a reader.

RESPONSE:

We do not have any evidence of false negatives and false positives in this study. However, we wanted to raise the possibility of false negatives and false positives, based on influences from various environmental factors and process as documented in literature. We agree the sentence is misleading to readers, and therefore should be clarified.

The sentence is revised as “It is important to understand eDNA behaviour spatially and temporary in a natural environment, with care in interpretation of eDNA result.”

Literature Cited:

Scientific names should be italicized in the references, see #11, 24, 47.

RESPONSE:

Revised as suggested.

Figures:

Figure 1 and 2 were blurry and difficult to read. Not sure if this was a conversion issue within the submission stage.

I would have liked to see the correlation figures and not the raw data.

RESPONSE:

We believed the resolution of figures maybe a conversion problem during submission. We will submit a better quality of figures in re-submission. We tested making a correlation figure, but it did not seem so useful and meaningful because there were so many samples with zero amplification, so we kept Figure 2 as is.

---

## [Decision Letter · Decision Letter 1]

29 Sep 2021

PONE-D-21-13336R1Using eDNA techniques to find the endangered big-headed turtles (Platysternon megacephalum)PLOS ONE

Dear Dr. FONG,

Thank you for submitting your manuscript to PLOS ONE. After careful consideration, we feel that it has merit but does not fully meet PLOS ONE’s publication criteria as it currently stands. Therefore, we invite you to submit a revised version of the manuscript that addresses the points raised during the review process.

We look forward to receiving your revised manuscript.

Kind regards,

Mark A. Davis, Ph.D.

Academic Editor

PLOS ONE

Journal Requirements:

Additional Editor Comments:

Thank you for submitting your revised manuscript to PLOS one. Two reviewers again reviewed the manuscript, and are in agreement that edits made have substantially improved this manuscript. In their thorough reviews, each provides a number of cosmetic recommendations to improve the manuscript. In addition, a few substantive changes are encouraged. First, both reviewers agree, and the AE concurs, that there remains some confusion around the LoD and LoQ calculations and presentation. As this information is critical for end-users who might consider adopting the assay to fully vet the assay, these issues require adjudication. Relatedly, reviewer 1 recommends (and the AE concurs) that completing the Thalinger et al. (2021) assay validation checklist and including as a supplementary file is necessary. This is incredibly valuable to the eDNA research community, and should always accompany the publication of a novel assay. Finally, Reviewer #2 expresses some concern about the modeling design, parameterization, results, and interpretation that requires careful consideration.

Reviewers' comments:

Reviewer's Responses to Questions

**Comments to the Author**

1. If the authors have adequately addressed your comments raised in a previous round of review and you feel that this manuscript is now acceptable for publication, you may indicate that here to bypass the “Comments to the Author” section, enter your conflict of interest statement in the “Confidential to Editor” section, and submit your "Accept" recommendation.

Reviewer #1: (No Response)

Reviewer #2: All comments have been addressed

2. Is the manuscript technically sound, and do the data support the conclusions?

Reviewer #1: Yes

Reviewer #2: Yes

3. Has the statistical analysis been performed appropriately and rigorously? 

Reviewer #1: Yes

Reviewer #2: I Don't Know

4. Have the authors made all data underlying the findings in their manuscript fully available?

Reviewer #1: Yes

Reviewer #2: Yes

5. Is the manuscript presented in an intelligible fashion and written in standard English?

Reviewer #1: Yes

Reviewer #2: Yes

6. Review Comments to the Author

Reviewer #1: General comments

I have reviewed the manuscript ‘Using eDNA techniques to find the endangered big-headed turtles (Platysternon megacephalum)’. I am largely satisfied with the changes the authors have made to the manuscript, but still have a number of minor comments I would like the authors to address to help improve the clarity of the manuscript. I have detailed these in the specific comments to the authors below. I also recommend that the authors complete the validation checklist provided in Thalinger et al. (2021) for the assay used and provide this as a supplementary file.

Thalinger, B., Deiner, K., Harper, L.R., Rees, H.C., Blackman, R.C., Sint, D., Traugott, M., Goldberg, C.S. & Bruce, K. (2021) A validation scale to determine the readiness of environmental DNA assays for routine species monitoring. Environmental DNA, 3, 823–836.

https://doi.org/10.1002/edn3.189

Specific comments

Line 19: Change ‘survey’ to ‘surveyed’.

Lines 22-23: Change ‘eDNA quantity and the environmental variables tested had no significant 23 relationship’ to ‘There was no significant relationship between eDNA quantity and the environmental variables tested.

Line 26: Remove comma.

Line 81: Remove comma before ‘using’

Line 89: Insert ‘to’ after ‘Prior’. How long was equipment immersed in 10% bleach solution for sterilisation?

Line 96: Insert ‘the’ before ‘fluorometer’.

Line 99: Change ‘the process’ to ‘sample processing’.

Line 100: Insert ‘23’ before ‘extractions’.

Line 107: Remove ‘a’ before ‘reliable’.

Lines 108-111: I appreciate that the authors will not be repeating qPCR at this stage due to cost and time, but one way of increasing assay sensitivity would have been to use larger reaction volumes for qPCR.

Lines 111-112: What volume was used for the TaqMan Exogenous Internal Positive Control assay in qPCR reactions?

Line 128: Remove comma after ‘uncertain’.

Line 129: Change ‘of’ to ‘performed on’.

Line 147: Change ‘eDNA water samples’ to ‘water samples for eDNA analysis’. Insert ‘walked’ before ‘along’.

Line 153: Insert ‘(‘ before ‘1)’.

Lines 181-183: I suggest joining these sentences with the previous paragraph.

Lines 191-190: Change ‘all negative controls exhibited no qPCR fluorescence signal’ to ‘none of the negative controls exhibited qPCR fluorescence signal’. There is a slight issue here as the qPCR standard curve used in the present study only went down to 100 copies/ul. This is the limit of quantification in this study as 100 copies/ul was the last point on the standard curve against which eDNA samples were being compared. If samples quantified below this, you can only somewhat rely on the copy number estimates produced for these samples as they are outwith the range of the standard curve. It is not good practice to use a standard curve from other studies (e.g. Lam et al. 2020) as it does not accurately reflect the qPCR reactions performed in the present study. I suggest the authors clarify whether their samples were above or below 100 copies/ul. If below, they can still count these as positive detections as there is clearly eDNA present, but these samples shouldn’t really be used in analyses examining eDNA quantity.

Line 243: Change ‘, but, a significant, positive’ to ‘, but a significant positive’.

Line 246: Change ‘in’ to ‘on’.

Line 248: Change ‘them was statistically significant in affecting’ to ‘these covariates significantly influenced’.

Line 276: Change ‘species specific’ to ‘species-specific’.

Line 293: Change ‘Degradation’ to ‘degradation’.

Line 298 and 308: Change ‘Production’ to ‘production’.

Line 310: Change ‘increases’ to ‘increase’.

Line 313: Change ‘produce less eDNA by shedding’ to ‘shed less eDNA’.

Line 323: Change ‘in hopes’ to ‘in the hope’.

Line 326: Change ‘Transport’ to ‘transport’.

Line 328: Change ‘which’ to ‘where’.

Line 329: Change ‘cause eDNA distribution vary’ to ‘can cause eDNA distribution to’.

Line 340: Change ‘Dilution’ to ‘dilution’.

Line 345: Change ‘amplifying’ to ‘resuspending’.

Line 347: Change ‘model’ to ‘models’.

Line 363: Change ‘result’ to ‘results’.

Line 368: Change ‘volume’ to ‘volumes’.

Line 375: Insert ‘of’ after ‘detection’.

Line 376: Change ‘present/absent’ to ‘presence/absence’.

Line 389: Insert ‘reaction volume’ before ‘and’.

Reviewer #2: The authors provided thoughtful and thorough revision of their paper. Great work!

However, I still have some confusion around reporting of assay performance and present minor comments below.

L89: Add “to” to this sentence: “Prior to each lab work step,..”

L147-148: Suggest revising sentence to read something like, “For active searching, we looked for P. megacephalum along the stream at night using headlamps.”

L149: Change “was” to “were”

L173: Delete extra “.” after “...and all KP streams. (n=8).”

Reporting of qPCR metrics (R2, %efficiency and LOD):

1. Because the authors quantify, both the limit of detection and limit of quantification need to be reported. So the limit of quantification also needs to be reported. Additionally, the copy number estimates need to be reported as reference to the LOD in the paper cited copies/μL.

2. I am confused about the multiple R2 and % efficiency reported, at L106 & L385 (R2=0.97, efficiency=94.15%), L189-190 and Supplemental Table 1 (R2=0.957 ± 0.03; efficiency: 96.99% ± 5.19), and the discussion of lower values at L386. I think reporting of the values only from the qPCR runs in this study should be reported in the results section and then mention the comparison with the previous study in the limitation sections (L381-390).

3. As a reader, I am also confused about the reported LOD at 10 copies/μL at L191. In the paper and in the authors’ response it looks like the limit of detection run in this study was 100 copies/μL (L116). I am not sure that LOD is necessarily transferable from study to study, especially given that the R2 and % efficiency for some runs were lower than what would be recommended to quantify eDNA.

Results:

L238-240: Add the rs and p values here for the correlation relationships between temperature and pH.

I had previously suggested that occupancy modeling might be useful in determining which factors are important to detect turtle eDNA. I sincerely apologize if this was not clear. But there is not enough explanation of the parameters included and the results are not fully explained. For example, use of AICc and p-values are generally not used together, there is no null model or global model included. As these relationships are still compared with correlations at L238-240, I suggest deleting this section (246-249), as it's not completely explained.

7. PLOS authors have the option to publish the peer review history of their article (what does this mean?). If published, this will include your full peer review and any attached files.

Reviewer #1: No

Reviewer #2: No

---

## [Author Response · Author response to Decision Letter 1]

11 Nov 2021

Reviewer #1: General comments

I have reviewed the manuscript ‘Using eDNA techniques to find the endangered big-headed turtles (Platysternon megacephalum)’. I am largely satisfied with the changes the authors have made to the manuscript, but still have a number of minor comments I would like the authors to address to help improve the clarity of the manuscript. I have detailed these in the specific comments to the authors below. I also recommend that the authors complete the validation checklist provided in Thalinger et al. (2021) for the assay used and provide this as a supplementary file.

Thalinger, B., Deiner, K., Harper, L.R., Rees, H.C., Blackman, R.C., Sint, D., Traugott, M., Goldberg, C.S. & Bruce, K. (2021) A validation scale to determine the readiness of environmental DNA assays for routine species monitoring. Environmental DNA, 3, 823–836.

https://doi.org/10.1002/edn3.189

RESPONSE:

Revised as suggested.

We appreciated the recommendation from the reviewer, and completed the validation checklist provided by the literature suggested. The checklist is attached as supplementary table 1.

Specific comments

Line 19: Change ‘survey’ to ‘surveyed’.

RESPONSE:

Revised as suggested.

Lines 22-23: Change ‘eDNA quantity and the environmental variables tested had no significant 23 relationship’ to ‘There was no significant relationship between eDNA quantity and the environmental variables tested.

RESPONSE:

Revised as suggested.

Line 26: Remove comma.

RESPONSE:

Revised as suggested.

Line 81: Remove comma before ‘using’

RESPONSE:

Revised as suggested.

Line 89: Insert ‘to’ after ‘Prior’. 

RESPONSE:

Revised as suggested. 

How long was equipment immersed in 10% bleach solution for sterilisation?

We immersed equipment in 10% bleach solution for at least 5 minutes for sterilization each time. We clarified the details in the corresponding sentences.

Line 96: Insert ‘the’ before ‘fluorometer’.

RESPONSE:

Revised as suggested.

Line 99: Change ‘the process’ to ‘sample processing’.

RESPONSE:

Revised as suggested.

Line 100: Insert ‘23’ before ‘extractions’.

RESPONSE:

Revised as suggested.

Line 107: Remove ‘a’ before ‘reliable’.

RESPONSE:

Revised as suggested.

Lines 108-111: I appreciate that the authors will not be repeating qPCR at this stage due to cost and time, but one way of increasing assay sensitivity would have been to use larger reaction volumes for qPCR.

RESPONSE:

Thank you for your suggestion, we will keep this in mind for the future.

Lines 111-112: What volume was used for the TaqMan Exogenous Internal Positive Control assay in qPCR reactions?

RESPONSE:

TaqMan Exogenous Internal Positive Control assay composed of two parts: IPC-Mix and IPC-DNA. Following manufacturer’s instruction, we added 1 μL of IPC-Mix and 0.2 μL of IPC-DNA into each reaction, We clarified the details in the corresponding sentences.

Line 128: Remove comma after ‘uncertain’.

RESPONSE:

Revised as suggested.

Line 129: Change ‘of’ to ‘performed on’.

RESPONSE:

Revised as suggested.

Line 147: Change ‘eDNA water samples’ to ‘water samples for eDNA analysis’. Insert ‘walked’ before ‘along’.

RESPONSE:

Revised as suggested.

Line 153: Insert ‘(‘ before ‘1)’.

RESPONSE:

Revised as suggested.

Lines 181-183: I suggest joining these sentences with the previous paragraph.

RESPONSE:

Revised as suggested.

Lines 191-190: Change ‘all negative controls exhibited no qPCR fluorescence signal’ to ‘none of the negative controls exhibited qPCR fluorescence signal’. 

RESPONSE:

Revised as suggested.

There is a slight issue here as the qPCR standard curve used in the present study only went down to 100 copies/ul. This is the limit of quantification in this study as 100 copies/ul was the last point on the standard curve against which eDNA samples were being compared. If samples quantified below this, you can only somewhat rely on the copy number estimates produced for these samples as they are outwith the range of the standard curve. It is not good practice to use a standard curve from other studies (e.g. Lam et al. 2020) as it does not accurately reflect the qPCR reactions performed in the present study. I suggest the authors clarify whether their samples were above or below 100 copies/ul. If below, they can still count these as positive detections as there is clearly eDNA present, but these samples shouldn’t really be used in analyses examining eDNA quantity.

RESPONSE:

We follow the suggestion from the reviewer, indicating whether the samples were above the limit of quantification (100 copies/μL) in text, and in Figure 2. We also remove samples with below 100 copies/μL from statistical analysis. Upon the removal of the data, eDNA quantity and turtle capture rate had no significant relationship anymore (rs = 0.267, p = 0.056). We updated the latest result in the Result section.

Line 243: Change ‘, but, a significant, positive’ to ‘, but a significant positive’.

RESPONSE:

Revised as suggested.

Line 246: Change ‘in’ to ‘on’.

RESPONSE:

Revised as suggested.

Line 248: Change ‘them was statistically significant in affecting’ to ‘these covariates significantly influenced’.

RESPONSE:

Revised as suggested.

Line 276: Change ‘species specific’ to ‘species-specific’.

RESPONSE:

Revised as suggested.

Line 293: Change ‘Degradation’ to ‘degradation’.

RESPONSE:

Revised as suggested.

Line 298 and 308: Change ‘Production’ to ‘production’.

RESPONSE:

Revised as suggested.

Line 310: Change ‘increases’ to ‘increase’.

RESPONSE:

Revised as suggested.

Line 313: Change ‘produce less eDNA by shedding’ to ‘shed less eDNA’.

RESPONSE:

Revised as suggested.

Line 323: Change ‘in hopes’ to ‘in the hope’.

RESPONSE:

Revised as suggested.

Line 326: Change ‘Transport’ to ‘transport’.

RESPONSE:

Revised as suggested.

Line 328: Change ‘which’ to ‘where’.

RESPONSE:

Revised as suggested.

Line 329: Change ‘cause eDNA distribution vary’ to ‘can cause eDNA distribution to’.

RESPONSE:

Revised as suggested.

Line 340: Change ‘Dilution’ to ‘dilution’.

RESPONSE:

Revised as suggested.

Line 345: Change ‘amplifying’ to ‘resuspending’.

RESPONSE:

Revised as suggested.

Line 347: Change ‘model’ to ‘models’.

RESPONSE:

Revised as suggested.

Line 363: Change ‘result’ to ‘results’.

RESPONSE:

Revised as suggested.

Line 368: Change ‘volume’ to ‘volumes’.

RESPONSE:

Revised as suggested.

Line 375: Insert ‘of’ after ‘detection’.

RESPONSE:

Revised as suggested.

Line 376: Change ‘present/absent’ to ‘presence/absence’.

RESPONSE:

Revised as suggested.

Line 389: Insert ‘reaction volume’ before ‘and’.

RESPONSE:

Revised as suggested. 

Reviewer #2: The authors provided thoughtful and thorough revision of their paper. Great work!

However, I still have some confusion around reporting of assay performance and present minor comments below.

L89: Add “to” to this sentence: “Prior to each lab work step,..”

RESPONSE:

Revised as suggested.

L147-148: Suggest revising sentence to read something like, “For active searching, we looked for P. megacephalum along the stream at night using headlamps.”

RESPONSE:

Revised as suggested.

L149: Change “was” to “were”

RESPONSE:

Revised as suggested.

L173: Delete extra “.” after “...and all KP streams. (n=8).”

RESPONSE:

Revised as suggested.

Reporting of qPCR metrics (R2, %efficiency and LOD):

1. Because the authors quantify, both the limit of detection and limit of quantification need to be reported. So the limit of quantification also needs to be reported. Additionally, the copy number estimates need to be reported as reference to the LOD in the paper cited copies/μL.

RESPONSE:

We apologize for the confusion between LOD and LOQ. In this study, we used 100 copies/μL as the lowest point of standard curve, hence, the limit of quantification. In Lam et al. (2020), the limit of detection was tested to be 10 copies/μL. We clarified the limit of detection in the Methodology section – Quantitative PCR, and the limit of quantification in the Methodology section - Exploration of eDNA Data.

2. I am confused about the multiple R2 and % efficiency reported, at L106 & L385 (R2=0.97, efficiency=94.15%), L189-190 and Supplemental Table 1 (R2=0.957 ± 0.03; efficiency: 96.99% ± 5.19), and the discussion of lower values at L386. I think reporting of the values only from the qPCR runs in this study should be reported in the results section and then mention the comparison with the previous study in the limitation sections (L381-390).

RESPONSE:

We apologize for making the confusion. We removed qPCR values (R2 and % efficiency) from other studies from the methodology and result section to minimize confusion. All comparison of qPCR values can be only found in Discussion – Limitation of the study sections as suggested. 

3. As a reader, I am also confused about the reported LOD at 10 copies/μL at L191. In the paper and in the authors’ response it looks like the limit of detection run in this study was 100 copies/μL (L116). I am not sure that LOD is necessarily transferable from study to study, especially given that the R2 and % efficiency for some runs were lower than what would be recommended to quantify eDNA.

RESPONSE:

To be conservative, we removed amplified eDNA samples with quantity below 100 copies/μL from the analysis. Upon the removal of the data, eDNA quantity and turtle capture rate had no significant relationship anymore (rs = 0.267, p = 0.056). We updated the latest result in the Result section.

Results:

L238-240: Add the rs and p values here for the correlation relationships between temperature and pH.

RESPONSE:

The corresponding rs and p values were added into the sentences suggested.

I had previously suggested that occupancy modeling might be useful in determining which factors are important to detect turtle eDNA. I sincerely apologize if this was not clear. But there is not enough explanation of the parameters included and the results are not fully explained. For example, use of AICc and p-values are generally not used together, there is no null model or global model included. As these relationships are still compared with correlations at L238-240, I suggest deleting this section (246-249), as it's not completely explained.

RESPONSE:

We sincerely apologize for not enough explanation on the parameters. However, as the reviewer said, the relationship was compared with correlations at L238-240, we followed the suggestion from the reviewer, to delete the section, and the corresponding paragraph in the Methodology section.

---

## [Decision Letter · Decision Letter 2]

6 Dec 2021

PONE-D-21-13336R2Using eDNA techniques to find the endangered big-headed turtles (Platysternon megacephalum)PLOS ONE

Dear Dr. FONG,

Thank you for submitting your manuscript to PLOS ONE. After careful consideration, we feel that it has merit but does not fully meet PLOS ONE’s publication criteria as it currently stands. Therefore, we invite you to submit a revised version of the manuscript that addresses the points raised during the review process.

The manuscript has improved with each iteration, and the authors are to be commended. This manuscript now meets PLOS One's publication criteria and can be accepted for publication pending responses to Reviewer 1's recommendations, particularly the final one regarding the Assay Validation Scale. Thank you for your outstanding work, and I look forward to seeing this manuscript published in the near future.

We look forward to receiving your revised manuscript.

Kind regards,

Mark A. Davis, Ph.D.

Academic Editor

PLOS ONE

Journal Requirements:

Additional Editor Comments (if provided):

Thank you for your careful consideration of the previous reviews. The same two reviewers who commented on previous versions reviewed the latest revision. Both reviewers agree and the AE concurs that the publication is largely acceptable for publication. That said, Reviewer 1 provides some critical revisions that need to occur, particularly with respect to the Thalinger et al. scale. I ask that you please make these edits, resubmit with a letter responding to Reviewer 1's recommendations. Once that is received, I will accept the manuscript for publication without further review.

Reviewers' comments:

Reviewer's Responses to Questions

**Comments to the Author**

1. If the authors have adequately addressed your comments raised in a previous round of review and you feel that this manuscript is now acceptable for publication, you may indicate that here to bypass the “Comments to the Author” section, enter your conflict of interest statement in the “Confidential to Editor” section, and submit your "Accept" recommendation.

Reviewer #1: (No Response)

Reviewer #2: All comments have been addressed

2. Is the manuscript technically sound, and do the data support the conclusions?

Reviewer #1: Yes

Reviewer #2: Yes

3. Has the statistical analysis been performed appropriately and rigorously? 

Reviewer #1: Yes

Reviewer #2: Yes

4. Have the authors made all data underlying the findings in their manuscript fully available?

Reviewer #1: Yes

Reviewer #2: Yes

5. Is the manuscript presented in an intelligible fashion and written in standard English?

Reviewer #1: Yes

Reviewer #2: Yes

6. Review Comments to the Author

Reviewer #1: General comments

I have reviewed the manuscript ‘Using eDNA techniques to find the endangered big-headed turtles (Platysternon megacephalum)’. I am largely satisfied with the changes the authors have made to the manuscript, and recommend the manuscript be published after minor comments are addressed. I have detailed these in the specific comments to the authors below.

Specific comments

Line 99: Add number of sampled in a set for DNA quantification, i.e. ‘one for each set of X samples for DNA quantification’.

Line 100: Change ‘2-3’ to ‘two or three’.

Line 107: Change ‘qPCR assay provided’ to ‘the selected qPCR assay’. Change ‘supplementary table’ to ‘Supplementary Table’.

Line 113: Change ‘2-3’ to ‘two or three’.

Line 114: Change ‘are’ to ‘were’.

Line 124: Change ‘2-5’ to ‘two to five’.

Lines 125-126: Change ‘when the qPCR assay recovered DNA amplification’ to ‘when qPCR amplification was observed’.

Line 127: Change ‘is’ to ‘was’.

Line 129: Change ‘performed on all qPCR reactions of the sample’ to ‘of all qPCR replicates performed on the sample’.

Line 133-134: Remove ‘were’. Change ‘had zero amplification’ to ‘did not amplify’.

Line 135: Change ‘2-3’ to ‘two or three’.

Lines 136 and 137: Change ‘showed amplification’ to ‘amplified.

Line 144: Change ‘1-2’ to ‘one to two’.

Line 155: Change ‘Shapiro-Wilk’s’ to ‘Shapiro-Wilk’.

Line 162: Change ‘quantity data’ to ‘concentration’.

Line 203: Change ‘2-5’ to ‘two to five’.

Line 220: Change ‘3 HR, 5 UN’ to ‘three HR, five UN’.

Line 223: Change ‘6 HR’ to ‘six HR’.

Line 237: Change ‘[‘ to ‘(‘.

Line 238: Remove ‘)’. Insert ‘:’ after ‘streams’.

Line 239: Change ‘]’ to ‘)’.

Line 256: Change ‘1-2’ to ‘one to two’.

Line 280: Change ‘in’ to ‘with’.

Line 281: Change ‘organism’ to ‘individuals’.

Line 360: Change ‘1-2L’ to ‘1-2 L’.

Line 378: Change ‘suggest’ to ‘suggests’.

Line 647: Change ‘of qPCR assay’ to ‘for the qPCR assay’.

Supplementary Table 1: The assay validation checklist from Thalinger et al. (2021) has not been completed properly. Please complete the ‘Checklist template’ sheet in the Excel spreadsheet titled ‘Appendix S1’ that was uploaded as part of the Supporting Information for Thalinger et al. (2021).

Thalinger, B., Deiner, K., Harper, L.R., Rees, H.C., Blackman, R.C., Sint, D., Traugott, M., Goldberg, C.S. & Bruce, K. (2021) A validation scale to determine the readiness of environmental DNA assays for routine species monitoring. Environmental DNA, 3, 823–836.

https://doi.org/10.1002/edn3.189

Reviewer #2: (No Response)

7. PLOS authors have the option to publish the peer review history of their article (what does this mean?). If published, this will include your full peer review and any attached files.

Reviewer #1: No

Reviewer #2: No

---

## [Author Response · Author response to Decision Letter 2]

7 Dec 2021

Reviewer #1: General comments

I have reviewed the manuscript ‘Using eDNA techniques to find the endangered big-headed turtles (Platysternon megacephalum)’. I am largely satisfied with the changes the authors have made to the manuscript, and recommend the manuscript be published after minor comments are addressed. I have detailed these in the specific comments to the authors below.

Specific comments

Line 99: Add number of sampled in a set for DNA quantification, i.e. ‘one for each set of X samples for DNA quantification’.

RESPONSE:

Revised as suggested.

Line 100: Change ‘2-3’ to ‘two or three’.

RESPONSE:

Revised as suggested.

Line 107: Change ‘qPCR assay provided’ to ‘the selected qPCR assay’. Change ‘supplementary table’ to ‘Supplementary Table’.

RESPONSE:

Revised as suggested.

Line 113: Change ‘2-3’ to ‘two or three’.

RESPONSE:

Revised as suggested.

Line 114: Change ‘are’ to ‘were’.

RESPONSE:

Revised as suggested.

Line 124: Change ‘2-5’ to ‘two to five’.

RESPONSE:

Revised as suggested.

Lines 125-126: Change ‘when the qPCR assay recovered DNA amplification’ to ‘when qPCR amplification was observed’.

RESPONSE:

Revised as suggested.

Line 127: Change ‘is’ to ‘was’.

RESPONSE:

Revised as suggested.

Line 129: Change ‘performed on all qPCR reactions of the sample’ to ‘of all qPCR replicates performed on the sample’.

RESPONSE:

Revised as suggested.

Line 133-134: Remove ‘were’. Change ‘had zero amplification’ to ‘did not amplify’.

RESPONSE:

Revised as suggested.

Line 135: Change ‘2-3’ to ‘two or three’.

RESPONSE:

Revised as suggested.

Lines 136 and 137: Change ‘showed amplification’ to ‘amplified.

RESPONSE:

Revised as suggested.

Line 144: Change ‘1-2’ to ‘one to two’.

RESPONSE:

Revised as suggested.

Line 155: Change ‘Shapiro-Wilk’s’ to ‘Shapiro-Wilk’.

RESPONSE:

Revised as suggested.

Line 162: Change ‘quantity data’ to ‘concentration’.

RESPONSE:

Revised as suggested.

Line 203: Change ‘2-5’ to ‘two to five’.

RESPONSE:

Revised as suggested.

Line 220: Change ‘3 HR, 5 UN’ to ‘three HR, five UN’.

RESPONSE:

Revised as suggested.

Line 223: Change ‘6 HR’ to ‘six HR’.

RESPONSE:

Revised as suggested.

Line 237: Change ‘[‘ to ‘(‘.

RESPONSE:

Revised as suggested.

Line 238: Remove ‘)’. Insert ‘:’ after ‘streams’.

RESPONSE:

Revised as suggested.

Line 239: Change ‘]’ to ‘)’.

RESPONSE:

Revised as suggested.

Line 256: Change ‘1-2’ to ‘one to two’.

RESPONSE:

Revised as suggested.

Line 280: Change ‘in’ to ‘with’.

RESPONSE:

Revised as suggested.

Line 281: Change ‘organism’ to ‘individuals’.

RESPONSE:

Revised as suggested.

Line 360: Change ‘1-2L’ to ‘1-2 L’.

RESPONSE:

Revised as suggested.

Line 378: Change ‘suggest’ to ‘suggests’.

RESPONSE:

Revised as suggested.

Line 647: Change ‘of qPCR assay’ to ‘for the qPCR assay’.

RESPONSE:

Revised as suggested.

Supplementary Table 1: The assay validation checklist from Thalinger et al. (2021) has not been completed properly. Please complete the ‘Checklist template’ sheet in the Excel spreadsheet titled ‘Appendix S1’ that was uploaded as part of the Supporting Information for Thalinger et al. (2021).

Thalinger, B., Deiner, K., Harper, L.R., Rees, H.C., Blackman, R.C., Sint, D., Traugott, M., Goldberg, C.S. & Bruce, K. (2021) A validation scale to determine the readiness of environmental DNA assays for routine species monitoring. Environmental DNA, 3, 823–836.

https://doi.org/10.1002/edn3.189

RESPONSE:

Revised as suggested. An updated checklist as Supplementary Table 1 will be submitted with the manuscript.

Reviewer #2: (No Response)

---

## [Editor Report · Decision Letter 3]

16 Dec 2021

Using eDNA techniques to find the endangered big-headed turtles (Platysternon megacephalum)

PONE-D-21-13336R3

Dear Dr. FONG,

We’re pleased to inform you that your manuscript has been judged scientifically suitable for publication and will be formally accepted for publication once it meets all outstanding technical requirements.

Kind regards,

Mark A. Davis, Ph.D.

Academic Editor

PLOS ONE

Additional Editor Comments (optional):

Thank you for your most recent edits. I appreciate all the work you have done to address reviewer comments. I look forward to seeing this manuscript published. Please note that during the proof phase, you should correct the title to read "Using eDNA techniques to find the endangered big headed turtle..." (deleting the superfluous "s" at the end of "turtle")
---

## [Editor Report · Acceptance letter]

28 Jan 2022

PONE-D-21-13336R3 

Using eDNA techniques to find the endangered big-headed turtle (*Platysternon megacephalum*) 

Dear Dr. FONG:

I'm pleased to inform you that your manuscript has been deemed suitable for publication in PLOS ONE. Congratulations! Your manuscript is now with our production department. 

Kind regards, 

on behalf of

Dr. Mark A. Davis 

Academic Editor

PLOS ONE